# Female peer mentors early in college have lasting positive impacts on female engineering students that persist beyond graduation

Deborah J. Wu [1], Kelsey C. Thiem [2] & Nilanjana Dasgupta [3] ✉

Expanding the talent pipeline of students from underrepresented backgrounds in STEM has been a priority in the United States for decades. However, potential solutions to increase the number of such students in STEM academic pathways, measured using longitudinal randomized controlled trials in real-world contexts, have been limited. Here, we expand on an earlier investigation that reported results from a longitudinal field experiment in which undergraduate female students ($N = 150$) interested in engineering at college entry were randomly assigned a female peer mentor in engineering, a male peer mentor in engineering, or not assigned a mentor for their first year of college. While an earlier article presented findings from participants' first two years of college, the current article reports the same participants' academic experiences for each year in college through college graduation and one year post-graduation. Compared to the male peer mentor and no mentor condition, having a female peer mentor was associated with a significant improvement in participants' psychological experiences in engineering, aspirations to pursue postgraduate engineering degrees, and emotional well-being. It was also associated with participants' success in securing engineering internships and retention in STEM majors through college graduation. In sum, a low-cost, short peer mentoring intervention demonstrates benefits in promoting female students' success in engineering from college entry, through one-year post-graduation.

Student engagement, success, persistence, and career pursuit in science, technology, engineering, and mathematics (STEM) is a high priority in the United States, particularly given the demand for a skilled STEM workforce and the short supply of college graduates with skills and interest in STEM careers[1,2]. The shortage of skilled STEM workers, especially in engineering, is exacerbated by the underrepresentation of females and racial ethnic minorities[3,4]. Sex and race gaps in STEM participation may also exacerbate income inequality between the sexes, and between racial minorities and white people, given relatively higher salaries in STEM professions compared to many other professions[5,6].

A growing body of research has been using randomized controlled trials to test whether social psychological interventions can reduce group-based disparities in students' STEM participation and performance. Many of these interventions train individuals to cognitively reappraise their experience, with an emphasis on teaching underrepresented individuals to mentally adapt to their environment (we call these mental reappraisal interventions)[7–16]. Using reappraisal strategies such as self-affirmation, growth mindset, and emotion regulation have been successful in reducing sex, race, and class disparities in academic performance[7–16]. For instance, brief affirmation interventions in which students reflect on personally held values have been

[1]Department of Psychology, Northwestern University, Evanston, IL 60201, USA. [2]Department of Counseling Psychology, Social Psychology, and Counseling, Ball State University, Muncie, IN 47306, USA. [3]Department of Psychological and Brain Sciences, University of Massachusetts Amherst, Amherst, MA 01003, USA. ✉e-mail: nd@umass.edu

shown to decrease sex and race gaps in academic performance in both lab and field settings[7–12]. Prompting students to cultivate a growth mindset (i.e., viewing intelligence or performance as changeable rather than a fixed ability) improves grades among previously low-performing students[13–15]. Reappraising anxiety as a positive mental state also decreases the achievement gap in grades between high and low-income students[16]. These interventions make important contributions toward academic equity, but they also have limitations. First, by focusing on mental reappraisal levers of change, these studies place the responsibility on underrepresented individuals to adjust to academic institutions by changing their mindset rather than placing the responsibility on academic institutions to change learning environments and meet the needs of diverse students. Second, several of these studies focus on grades and test performance as measures of interest but do not assess students' subjective experiences in academic spaces (e.g., feelings of belonging, confidence, anxiety, motivation)[8,10,12–16]. Emphasis on the subjective experiences of students from underrepresented backgrounds is critical because this may affect persistence and interest even in the absence of objective performance gaps[17,18]. Third, much of the research investigating mental reappraisal interventions did not target STEM education specifically, but instead focused on general academic performance (cumulative grade point average), and many of these studies were conducted with adolescents and not college students[7–16].

Another body of research shines light on the impacts of increasing the visibility of successful own-group role models for underrepresented students in STEM. This research testing the power of role models in STEM is consistent with the Stereotype Inoculation Model, which posits that exposure to successful own-group experts and peers protects one's mind against noxious stereotypes that cast doubt on one's ability[18]. This research shows that reading about, or interacting with, successful own-group role models closes sex and race gaps by increasing test performance and course grades for females and racial ethnic minorities. Moreover, exposure to these role models also bolsters feelings of confidence, belonging, and STEM identification[17,19–25]. Collectively, this work points to evidence-based solutions that promote equity in STEM education, both in terms of objective metrics (such as test performance and grades) and subjective metrics (confidence, motivation, and belonging), thereby amplifying the importance of visible representation of underrepresented groups in STEM environments.

Yet this research also has limitations, as these studies typically assess how role models impact student outcomes at a single time-point or over a short period of time (e.g., up to two weeks after interacting with the role model), providing insufficient information about long-term impacts[17,19–25] Moreover, the majority of these studies were laboratory experiments conducted in artificial settings rather than naturally existing environments[19–24], which means participants did not have opportunities to form what could be considered authentic mentoring relationships with role models. In our view, authentic long-term mentoring relationships differ from brief exposure to, or fleeting interactions, with role models. Authentic mentoring relationships allow mentees to seek advice from role models, get assistance when they struggle, and expand their academic and professional networks through their role model or mentor's contacts. Indeed, past research shows that when students have authentic mentoring relationships with faculty mentors in STEM, their academic outcomes are enhanced. For example, engineering students who have authentic mentoring relationships with faculty members (in which faculty members provide personalized attention and mentoring to the student), are more likely to persist and perform well in engineering classes[26]. Black students at historically Black universities are more likely to work directly with faculty when they struggle academically[27] and report higher well-being[28,29] as compared to Black students at predominantly white universities who receive less one-on-one attention from faculty[30].

Collectively, extant research suggests that authentic mentoring relationships between students and mentors who share their mentees' marginalized identity are likely to be more impactful compared to brief exposure to such individuals[29,31].

Using the Stereotype Inoculation Model[18] as our theoretical framework, the current research predicts that opportunities to form authentic mentoring relationships with role models who share one's marginalized identity will have substantial benefits for mentees by reinforcing their confidence, motivation, and abilities over time. Using a randomized controlled trial, we examine whether and how peer mentoring relationships in engineering in the first year of college influence female engineering students' subjective experiences and objective academic outcomes from college entry through graduation and up to one year post-graduation.

This current study expands upon our previously published work (Dennehy & Dasgupta, 2017[32]) which examined the impact of peer mentorship on mentees' experiences in the first year of college when mentoring was active. This earlier investigation found that being randomly assigned a female peer mentor (as compared to a male mentor or no mentor) produced academic benefits for female students in the first year of college. Compared to their baseline measures (i.e., before classes began), female students assigned to a male mentor or no mentor increased in their feelings of anxiety relative to motivation in their engineering classes, and declined in their belonging, confidence in engineering, and aspirations to pursue post-graduate degrees in engineering during their first year of college. In contrast, participants assigned to female mentors showed no change in anxiety relative to motivation, and stable belonging, confidence, and post-college aspirations in engineering. These benefits endured for a second year for the 52% of the sample (78 out of 150 students) that had completed their second year of college at the time of our analysis. The remaining 48% of the sample had not completed their second year when the original paper was published.

The present research makes three contributions beyond the original 2017 study. First, by following the full sample of female participants for 3–5 years beyond the intervention year through Bachelor's degree completion plus one year post-graduation, we test if the impact of peer mentoring endures beyond Bachelor's degree attainment long after mentoring has ended. Second, the present research measures additional variables that capture work experience (engineering internships) and emotional well-being. We expect that emotional well-being may be a key protection against attrition from STEM given past research showing that minority status and social identity threat predicts daily experiences of emotional exhaustion and psychological burnout among females in STEM[33,34]. Third, the present study provides a more accurate measure of STEM retention than the 2017 paper, in which retention was measured at the end of the first year of college, well before the university deadline to make final decisions about academic majors. At most American universities, students are required to declare their major by the end of the second year of college and they often switch majors between the first and second year of college. By using official university transcripts to measure students' majors at college graduation, the current article provides a more accurate picture of how first-year peer mentorship impacts students' final decisions about academic majors, which has lasting impacts on their post-graduate career trajectory.

Our focus is on females in engineering in particular because compared to many other STEM majors, females comprise a smaller minority in engineering degree programs (21%)[35] and the workforce (13%)[36], despite being 51% of the American population[37] and 60% of college students at American universities[38]. Consistent with the Stereotype Inoculation Model[18] and prior research[17,19–25], we predicted that relationships with female peer mentors in the first year of college would act as social vaccines for young female students entering engineering, helping them disregard negative stereotypes, preserve

their motivation in engineering courses, and increase persistence and professional development in STEM. Importantly, we predicted these benefits would endure through one year post-graduation, 3 to 5 years after the conclusion of the peer mentorship intervention.

The present work also extends existing research on mentorship in several ways. First, although mentorship is a popular intervention commonly discussed in academia, government, and industry[39–43], many previous field-based mentorship initiatives are not controlled experiments, making it impossible to determine whether mentorship is the causal factor responsible for downstream outcomes[40]. For instance, in studies where participants self-selected into mentorship programs, the success of mentorship interventions could be due to pre-existing individual differences between people who self-selected into mentoring programs and those who did not, rather than the content of the mentorship program itself. Further, in other studies where mentees had the freedom to choose their own mentors, the success of mentorship interventions could be due to the special relationship between the mentor and mentee rather than mentorship in general. To resolve these causal inference problems, we conducted a randomized controlled field experiment[32] in which participants were unaware that the study was related to mentoring.

Second, our field experiment complements past research on role models. While in theory, role model interventions should create more supportive environments for underrepresented individuals, as noted earlier, most prior role model experiments used artificial lab settings (e.g., reading about successful role models) and did not create authentic relationships[19–24]. In contrast, our year-long field intervention investigated whether and how fostering authentic relationships with peer mentors would change female students' lived experiences in engineering, scaffold their success during a critical transition period into college, and sustain that success through the next 3–5 years, after their mentors have left the university. Furthermore, previous interventions addressing STEM disparities assessed impacts at a single time-point or over a relatively short period of time[8,9,16,17,19–25], providing insufficient information about long-term impacts. Although some research has demonstrated longer-term benefits of academic interventions (i.e., 2–3 years), these studies were not specific to STEM[7,11].

Finally, our research examines whether mentees gain benefits from having a mentor who does not share their social identity by testing whether female engineering students benefit from male mentorship. Previous research suggests that females in STEM can benefit from effective male mentors[23,24]. However, because these studies were conducted as short-term experiments using experimenters posing as role models, it is unclear whether these findings will generalize to real-world academic settings. Thus, we examined the long-term impacts of male mentorship for female students in engineering.

We conducted a field-based longitudinal randomized controlled experiment that examined female students' subjective and objective academic outcomes and compared the impacts of own-sex and other-sex peer mentors to a control condition. We recruited female undergraduates who were first-year and transfer students ($N = 150$) at a large public university who planned to major in engineering. Participants were randomly assigned a female peer mentor ($N = 52$), a male peer mentor ($N = 51$), or no mentor ($N = 47$) for 1 year. Peer mentors were junior or senior undergraduate students who volunteered to be mentors and were trained prior to being matched with mentees. Researchers described the study to mentors and mentees as aiming to identify barriers and opportunities experienced by college students in engineering. This generic description served two purposes. First, it ensured that participants who volunteered for this experiment were not opting in because of an interest in mentoring. Furthermore, it kept mentors and mentees blind to the role of sex in this study (e.g., mentors did not know that all mentees were female; neither mentors nor mentees knew that mentor sex was relevant to hypotheses).

Mentor-mentee dyads met several times (median = 4 meetings) during mentees' first year in college, after which mentoring interactions ended. We surveyed mentees from college entry through Bachelor's degree completion to one year post-graduation. Measured variables included students' subjective experiences in engineering (motivation in engineering courses and confidence in overall engineering abilities); participation in engineering internships that provide technical work experience; persistence in engineering and STEM majors; aspirations to pursue post-graduate degrees in engineering; and emotional well-being. All reported variables except for persistence in engineering and STEM majors were self-reported in surveys, while major-related persistence was measured using college transcripts. The first survey was administered before mentor assignment at the beginning of students' first year of college, serving as a baseline. Subsequent surveys were administered at the middle and end of the first year of college, and then once each subsequent year through college graduation plus one year post-college graduation. College transcripts were also obtained with student permission to get an objective record of participants' choice of major and grades.

## Results
### Data management and analysis
Results are described in two sections. First, we present results showing outcomes for which female students in engineering in our sample experienced significantly stronger benefits from female peer mentorship than male peer mentorship or no mentorship. Second, we identify which of these outcomes serves as a psychological mechanism that helps explain these differences.

For continuous dependent variables measured at multiple time-points (motivation in engineering courses, confidence in overall engineering abilities, aspirations to pursue post-graduate degrees, and emotional well-being), we investigated how participants' responses changed over time through college graduation and one-year post-graduation by coding the time-point at which each response was obtained. Time was centered at the beginning of year 1, before mentor assignment (baseline). Responses for subsequent time-points were scaled in reference to the month in which the survey responses were obtained relative to the baseline. The time variable was divided by 12, such that slope coefficients indicate yearly change.

We estimated multilevel models using Mplus 8[44], utilizing the full information maximum likelihood estimator, which is the recommended practice to handle missing data[45,46]. Each statistical model has two levels. Level 1 represents variables measured at multiple time-points within participants (motivation in engineering courses, confidence in overall engineering abilities, post-graduate aspirations, and emotional well-being). Level 2 represents the mentor condition—the independent variable that was systematically manipulated between participants. Each model tested whether mentor condition influenced change over time on the dependent variable. In level 1 of the model, the dependent variable was regressed on time, creating a slope representing change over time for that variable. Random effects were created for the intercept (participants' baseline responses at college entry) and slope (change over time for each participant). The intercept and slope were allowed to covary at level 2 to control for variance shared between the actual values on the dependent variable and its slope or change over time. At level 2, the intercept and slope of the dependent variable were both regressed on mentor condition. Given that mentor condition was a multi-categorical variable, two dummy codes were created to index the female mentor condition (1 = female mentor, 0 = male and no mentor conditions), male mentor condition (1 = male mentor, 0 = female and no mentor conditions), with the no mentor condition always being the reference group. We then tested the effect of mentor condition on longitudinal change over time on participants' experiences in two ways: (a) by examining whether the

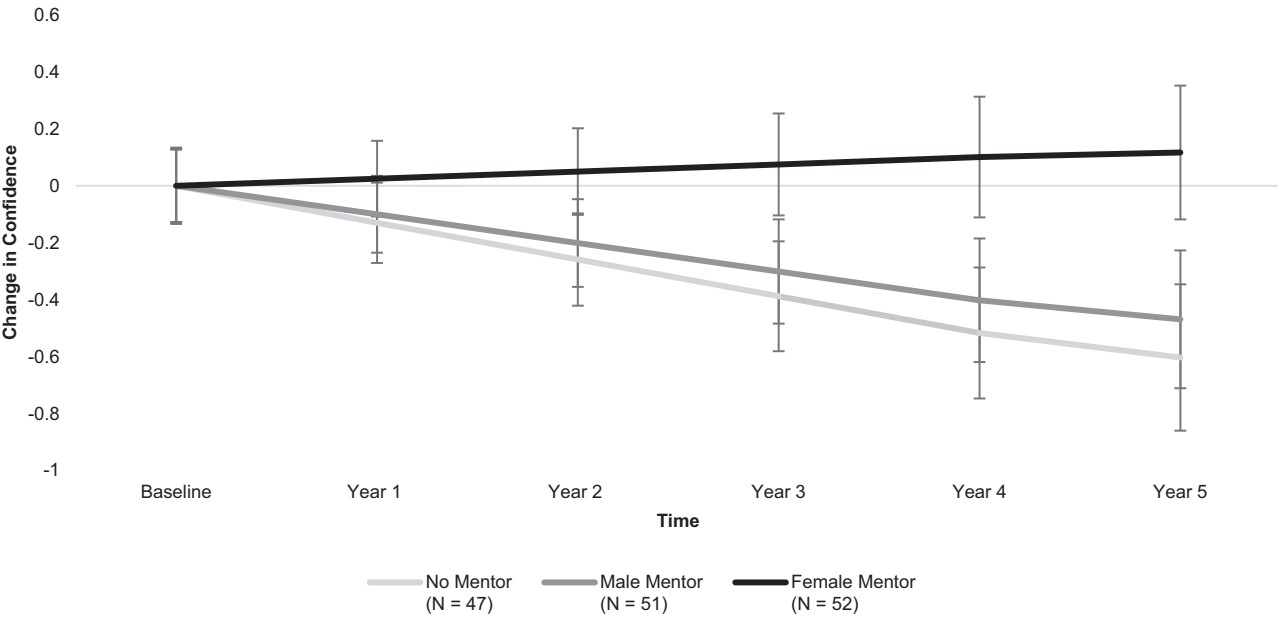

**Fig. 1 | Participants' confidence in engineering from college entry through one year post-graduation.** Depicts differences between intervention conditions (no mentor, male mentor, female mentor) in participants' average change in confidence in their engineering abilities over five years, relative to their confidence levels at the beginning of college, before mentors were assigned. Error bars designate +/−1 standard error of each condition's estimated mean confidence levels at that time-point. $N = 150$ participants.

trajectory of each mentor condition significantly changed from college entry through graduation to one-year post-graduation; and (b) by testing if these change trajectories significantly differed between mentor conditions. Change trajectories and the condition differences in change trajectories are reported using unstandardized regression values, corresponding to the change in the variable over one year and differences in these changes between conditions, respectively. Two-tailed significance values are reported. Furthermore, the multilevel models can be summarized by the following equation:

$$Y_{ij} = \beta_{00} + \beta_{01}\text{FemaleMentor} + \beta_{02}\text{MaleMentor} + \beta_{10}\text{Time} \\ + \beta_{11}\text{FemaleMentor*Time} + \beta_{12}\text{MaleMentor*Time} + r_{0i} \quad (1) \\ + r_{1i}Time + e_{ti}.$$

Post hoc Monte Carlo simulation analyses with 1000 simulations were also conducted for each effect, indicating the proportion of simulations in which the effect was significant[47]. We report power analyses for each significant effect. Dichotomous variables (participation in engineering internships, college major at graduation) were analyzed using chi-square tests to examine percentage differences between mentor conditions.

### Assignment to a female mentor was associated with improvement in participants' subjective experiences in engineering, objective choices in engineering, and emotional well-being

Whereas female participants without mentors ($B = -0.13$, $SE = 0.05$, $p = 0.007$, power = 0.92) and participants with male mentors ($B = -0.10$, $SE = 0.05$, $p = 0.025$, power = 0.77) showed a decline in their overall confidence in engineering from college entry through graduation and one year post-college, those with female mentors ($B = 0.03$, $SE = 0.04$, $p = 0.559$) maintained general confidence in engineering across time without decline. A contrast analysis revealed that the change trajectory of the female mentor condition was significantly different from the change trajectories of the no mentor and male mentor conditions ($B = 0.14$, $SE = 0.06$, $p = 0.010$, power = 0.94),

suggesting that female peer mentorship was associated with maintenance of confidence among participants across college and post-graduation. Upon comparing the trajectories of each condition separately, we found that participants without mentors ($B = 0.15$, $SE = 0.07$, $p = 0.017$, power = 0.81) and participants with male mentors ($B = 0.13$, $SE = 0.06$, $p = 0.045$, power = 0.66) reported greater decline in confidence versus those with female mentors. There was no difference between the change trajectories of the no mentor versus male mentor conditions ($B = 0.03$, $SE = 0.07$, $p = 0.660$). See Fig. 1.

Whereas participants without peer mentors ($B = -0.12$, $SE = 0.04$, $p = 0.006$, power = 0.92) and those assigned male mentors ($B = -0.14$, $SE = 0.04$, $p = 0.001$, power = 0.98) showed a significant decrease in motivation in their engineering classes from college entry through graduation to one year post-graduation, participants assigned female mentors showed no change in motivation ($B = -0.01$, $SE = 0.04$, $p = .782$). The change trajectory for female participants with female mentors was significantly different from the change trajectories in the other two conditions ($B = 0.12$, $SE = 0.05$, $p = 0.017$, power = 0.91), suggesting that relationships with female mentors preserved mentees' motivation in their engineering courses. When comparing conditions separately, the change trajectory for participants with female mentors versus male mentors was significantly different ($B = 0.13$, $SE = 0.06$, $p = 0.024$, power = 0.74). The change trajectory for those with no mentors fell in-between the other two conditions and did not significantly differ from either mentor condition (male mentor condition: $B = 0.02$, $SE = 0.06$, $p = 0.746$; female mentor condition: $B = 0.11$, $SE = 0.06$, $p = 0.064$). See Fig. 2.

Whereas 61% of participants without peer mentors and 65% of those with male mentors self-reported participating in engineering internships while in college, the proportion of internship participation was substantially higher (82%) for female participants with a female mentor (see Table 1). The difference between the female mentor condition and the other two conditions was significant ($\chi^2(1, N = 125) = 4.79$, $p = 0.029$). When comparing conditions separately, the percentage of female participants with female mentors who had participated in engineering internships was significantly greater than the

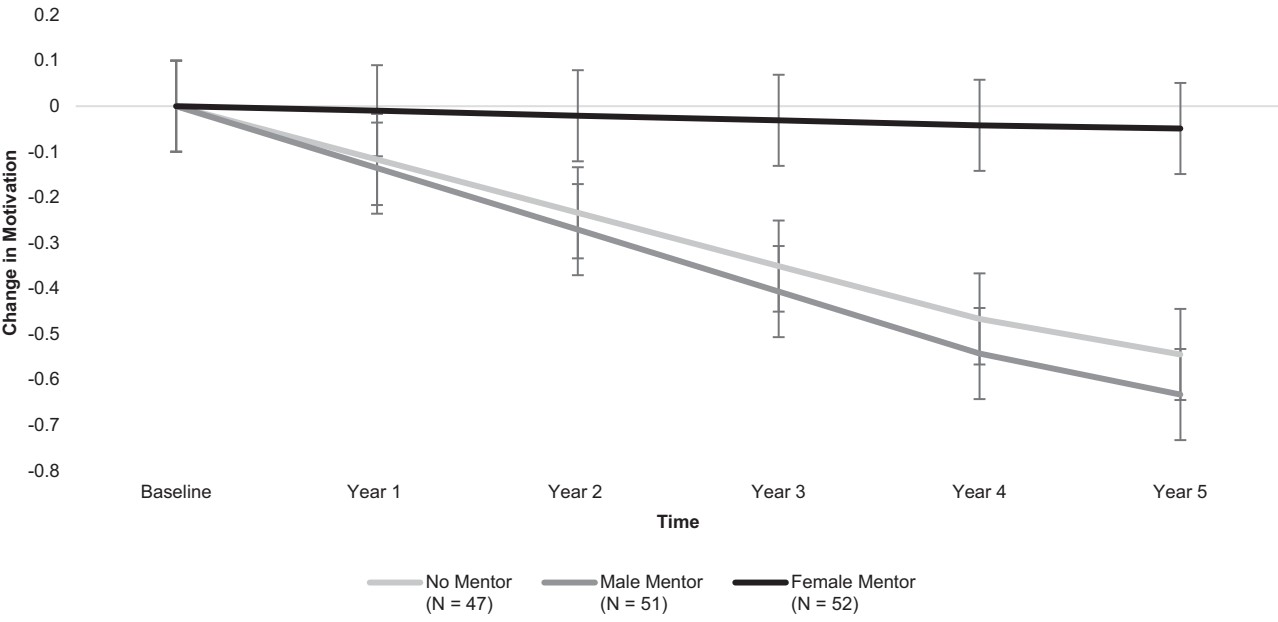

**Fig. 2 | Participants' motivation in engineering from college entry through one year post-graduation.** Depicts differences between intervention conditions (no mentor, male mentor, female mentor) in participants' average change in motivation in their engineering courses over five years, relative to their motivation levels at the beginning of college, before mentors were assigned. Error bars designate +/−1 standard error of each condition's estimated mean motivation levels at that time-point. $N = 150$ participants.

**Table 1 | Percentage (by Condition) that obtained an engineering internship, engineering degree, and a STEM degree**

| | No Mentor Condition | Male Mentor Condition | Female Mentor Condition |
|---|---|---|---|
| Engineering Internship | 61% | 65% | 82% |
| Engineering Degree | 66% | 71% | 79% |
| STEM Degree | 81% | 78% | 92% |

Table 1 depicts the percentage of participants (separated by condition) who obtained engineering internships at some point in college, as well as the percentage of participants who obtained a Bachelor's degree in engineering in particular or STEM in general.

percentage in the no mentor condition ($\chi^2(1, N = 82) = 4.58$, $p = 0.032$). The percentage in the male mentor condition did not significantly differ from the no mentor condition ($\chi^2(1, N = 81) = 0.18$, $p = 0.669$) or the female mentor condition ($\chi^2(1, N = 87) = 3.12$, $p = 0.077$).

We examined whether peer mentorship in the first year of college affected the proportion of students who graduated with an engineering major or any STEM major 2–4 years later (see Table 1). Whereas 66% of those with no mentor and 71% of those assigned a male mentor earned Bachelor's degrees in engineering, 79% of participants with a female mentor earned engineering degrees. There was not a statistically significant difference between conditions ($\chi^2(1, N = 150) = 1.85$, $p = 0.174$). However, for STEM majors (specifically, degrees in physical sciences, biological sciences, computer science, engineering, or mathematics[48]) a significant difference emerged by mentor condition: 92% of participants assigned female mentors graduated with STEM majors compared to 78% of participants assigned male mentors and 81% without mentors ($\chi^2(1, N = 150) = 4.09$, $p = 0.043$). Although having female peer mentors in the first year of college did not significantly increase engineering degree completion, it did significantly increase STEM degree completion compared to the other two conditions. The percentage of female STEM graduates in the female mentor condition was greater than in the male mentor condition ($\chi^2(1, N = 103) = 3.99$, $p = 0.046$). The no mentor condition did not significantly differ from the female mentor condition ($\chi^2(1, N = 99) = 2.84$, $p = 0.092$) or the male mentor condition ($\chi^2(1, N = 98) = 0.09$, $p = 0.767$).

Whereas participants without mentors ($B = -0.50$, $SE = 0.14$, $p < 0.001$, power $> 0.99$) and those assigned male mentors ($B = -0.44$, $SE = 0.13$, $p = 0.001$, power $> 0.99$) showed declining interest in pursuing post-graduate degrees in engineering from college entry through graduation and one year post-graduation, those assigned female mentors did not show this decrease in intentions to pursue post-graduate degrees in engineering ($B = -0.16$, $SE = 0.11$, $p = 0.159$). The change trajectory for the female mentor condition was significantly different from the no mentor and male mentor conditions ($B = 0.31$, $SE = 0.15$, $p = 0.036$, power $= 0.90$), showing that having a female peer mentor in the first year of college was associated with less of a decline in participants' reported post-graduate aspirations in engineering. When comparing conditions separately, the female mentor change trajectory did not significantly differ from the no mentor change trajectory ($B = 0.34$, $SE = 0.18$, $p = 0.056$) or the male mentor slope ($B = 0.28$, $SE = 0.17$, $p = 0.097$). The male mentor slope did not significantly differ from the no mentor slope ($B = 0.06$, $SE = 0.18$, $p = 0.739$). See Fig. 3.

We also examined the impact of mentorship condition on emotional health and well-being throughout college. Whereas participants without mentors and those assigned male mentors declined in their emotional well-being over time (no mentor: $B = -0.32$, $SE = 0.17$, $p = 0.061$, power $= 0.70$; male mentor: $B = -0.42$, $SE = 0.14$, $p = 0.002$, power $= 0.87$), emotional well-being for participants with a female mentor held steady throughout college ($B = 0.20$, $SE = 0.14$, $p = 0.147$). The difference in change trajectories between the female mentor condition and the other two conditions was significant ($B = 0.57$, $SE = 0.18$, $p = 0.001$, power $= 0.97$). The female mentor change trajectory was significantly different from the no mentor change trajectory ($B = 0.52$, $SE = 0.22$, $p = 0.017$, power $= 0.78$) and the male mentor change trajectory ($B = 0.62$, $SE = 0.19$, $p = 0.001$, power $= 0.89$). The no mentor and male mentor conditions did not differ ($B = 0.10$, $SE = 0.22$, $p = 0.659$). See Fig. 4.

### Confidence in engineering skills mediates the effect of female peer mentors on academic choices in engineering

Multilevel mediation analyses were conducted to test whether the effect of female peer mentors on academic choices in engineering

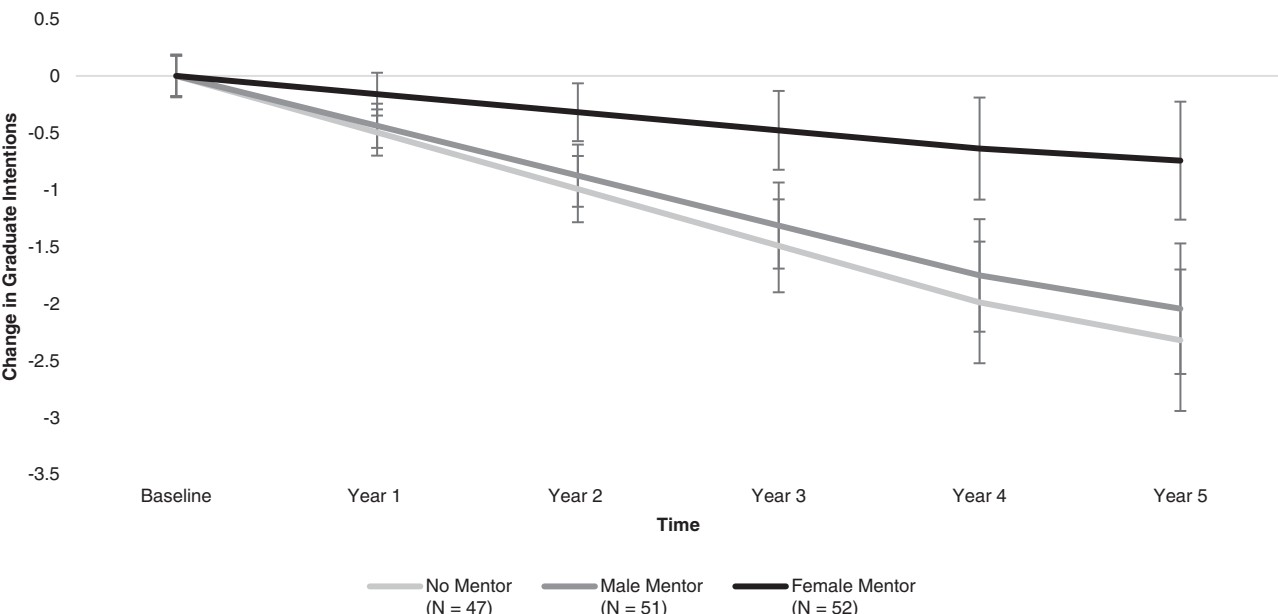

**Fig. 3 | Participants' intentions to pursue graduate school in engineering from college entry through one year post-graduation.** Depicts differences between intervention conditions (no mentor, male mentor, female mentor) in participants' average change in aspirations to pursue graduate degrees in engineering over five years, relative to their aspiration levels at the beginning of college, before mentors were assigned. Error bars designate +/−1 standard error of each condition's estimated mean aspiration levels at that time-point. $N = 150$ participants.

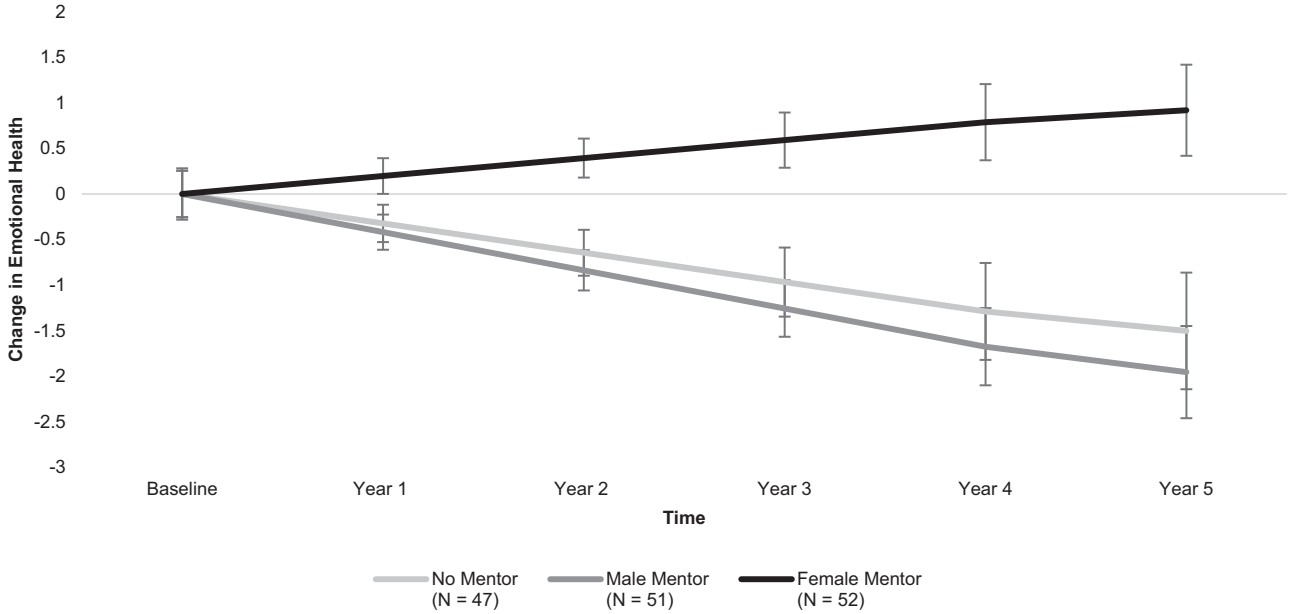

**Fig. 4 | Participants' emotional health and well-being from college entry through one year post-graduation.** Depicts differences between intervention conditions (no mentor, male mentor, female mentor) in participants' average change in emotional health over five years, relative to their reported levels during their first year of college. Error bars designate +/−1 standard error of each condition's estimated mean emotional health levels at that time-point. $N = 150$ participants.

(successfully securing engineering internships, majoring in STEM at graduation, and post-graduate aspirations in engineering) were explained by the change trajectories of confidence in overall engineering skills, motivation in engineering courses, or emotional well-being. For the objective outcomes (engineering internship and major at graduation), we used 2-1-2 mediation models[49] with mentor condition as the independent variable that varied between participants (level 2), change in confidence, motivation, and emotional well-being as the mediators that varied within participants (level 1), and

participation in engineering internships or graduating with a STEM degree as the dichotomous dependent variable that varied between participants (level 2). The mediation model for graduate aspirations was a 2-1-1 model because the outcome variable, graduate intentions, also varied within participants (level 1).

In each statistical model, all continuous variables (i.e., confidence, motivation, emotional well-being, graduate aspirations) were regressed on time, creating random effects for the intercepts and slopes. The dependent variable (internship, STEM degree, and slope of graduate

aspirations) was then regressed on the mediator slopes (change in confidence, motivation, and emotional well-being) and the mediator slopes were regressed on the independent variable (mentor condition). All continuous variables were centered at baseline to minimize multicollinearity and dichotomous variables were specified as categorical. In order to compare the female mentor condition with the other two conditions, condition was coded as a dichotomous variable (0 = no mentor and male mentor conditions and 1 = female mentor condition). Since Mplus does not allow for bootstrapping for indirect effects in a two-level model, each model was conducted with the Bayes estimator with 10,000 iterations, yielding near-identical results to bootstrapping[50].

A significant indirect effect indicated that the effect of the female mentor condition versus the other two conditions on participation in engineering internships was mediated through change in confidence ($B = 0.30$, 95% CI [0.04, 0.77]). Participants who had female mentors during their first year of college exhibited less decline in confidence in their engineering abilities compared to participants in the other two groups ($B = 0.15$, 95% CI [0.03, 0.28]). A smaller decline in confidence was associated with more success in securing engineering internships during college ($B = 2.13$, 95% CI [0.65, 3.84]). There were no significant indirect effects for motivation in engineering courses ($B = 0.12$, 95% CI [−0.08, 0.47]) or emotional well-being ($B = −0.01$, 95% CI [−0.17, 0.13]).

Both confidence ($B = 0.48$, 95% CI [0.06, 1.19]) and motivation ($B = 0.39$, 95% CI [0.03, 1.05]) significantly mediated the effect of the female mentor condition (versus the other two conditions) on graduating with STEM degrees. Compared to the other two conditions, having a female mentor in the first year of college reduced the decline in engineering confidence ($B = 0.16$, 95% CI [0.03, 0.28]) and motivation in engineering courses ($B = 0.15$, 95% CI [0.03, 0.27]), which in turn predicted increased likelihood of graduating with a STEM degree 3–5 years later (confidence: $B = 3.27$, 95% CI [1.31, 5.86]; motivation: $B = 2.80$, 95% CI [0.60, 5.52]). The indirect effect for emotional well-being was not significant ($B = 0.16$, 95% CI [−0.06, 0.71]).

Confidence in overall engineering abilities significantly mediated the effect of the female mentor condition on post-graduate aspirations in engineering ($B = 0.17$, 95% CI [0.02, 0.42]). Participants who had female mentors showed less decline in confidence compared to participants with male mentors or no mentors ($B = 0.15$, 95% CI [0.04, 0.25]), which in turn was associated with maintenance of self-reported post-graduate aspirations in engineering ($B = 1.18$, 95% CI [0.20, 2.28]). There were no significant indirect effects of motivation in engineering courses ($B = 0.07$, 95% CI [−0.09, 0.27]) or emotional well-being ($B = 0.01$, 95% CI [−0.05, 0.11]) on reported post-graduate aspirations in engineering.

## Discussion

Our results indicate that the benefits of same-sex peer mentoring relationships in the first year of college endure through the 3–5 year window of female students' college experience including one year post-graduation, long after mentorship has ended. These benefits emerge for both subjective experiences and objective academic outcomes. The durability of this intervention is notable, especially given its low-cost and light-touch nature. Mentees met with their mentors only four times on average during their first year of college.

Four findings are particularly noteworthy. First, whereas female participants assigned a male peer mentor or no mentor significantly declined in their confidence in overall engineering skills and their motivation in engineering courses, being assigned a female peer mentor was consistently associated with reduced decline in confidence and motivation. Second, in comparison to the other two conditions, being assigned a female peer mentor was associated with: increased rate of participation in engineering internships during college, increased graduation rate for Bachelor's degrees in STEM, and maintenance of higher aspirations to pursue post-graduate degrees in

engineering. Third, female mentorship resulted in greater reported mentee emotional well-being, whereas male mentorship or no mentorship was associated with a decline in emotional well-being through college and one-year post-college. Fourth, through mediation analyses, we found that stable confidence in one's engineering ability was the psychological mechanism among our measured outcomes that best explained why female peer mentorship promoted success in engineering internships, completion of STEM degrees, and pursuit of graduate training in engineering.

In addition to steadfast confidence at the individual level, we speculate about two additional theory-driven mechanisms—one that operates at an interpersonal level and another at an intrapersonal level—that may explain why female mentors were so effective. From an interpersonal perspective, female mentors may have expanded their mentees' access to valuable social networks to offset disparities in professional connections that often constrain members of minority groups[51,52]. Sociological research shows that for students who are minorities in an academic discipline, opportunities to develop a tight social network of peers who share their marginalized identity in that target profession increases professional success. For example, Yang and colleagues[53] found that female students in MBA programs were more successful in getting high-ranked jobs after graduation if they had a close network of female colleagues who advised them about where to apply, how to interview, and offered other background information about companies. Advice from close female connections (vs. male connections) may be especially useful because female informal advice-givers are cognizant of barriers about which male peers may not be aware. In Yang's study, female students had both males and females in their professional network, but female students who had a close network of trusted same-sex colleagues in their field were more successful in the job market. These social network findings may also apply to our study.

From an intrapersonal perspective, another potential mechanism may be that having a female mentor elicited a change in their mentees' mindset during a critical transitional period of their lives. As theorized by Walton[54], interventions may have a long-term impact if they change recursive processes early on. If an intervention is able to alter individuals' way of thinking and help them gain more constructive mindsets regarding their learning in the beginning of a new environment, they will be more likely to persist in their education. In contrast, if students remain caught in cycles of psychological threat and poor performance, they may be less likely to persist. Applying mindset change to our research, the benefit of female (as compared to male) mentors may have persisted because female students were able to develop constructive mindsets regarding engineering in their first year of college. Future research should directly examine these and other potential mechanisms driving the success of same-sex mentorship interventions that have long-lasting effects.

We also note limitations to our study due to its modest sample size and lack of information regarding our mentors. Females make up a small percentage of engineering students in the United States; at the time these data were collected, only 3.9% of female students entering the first year of college nationwide intended to major in engineering[33]. We collected data from four incoming first-year cohorts between 2011–2014 and followed their progress across the span of 8 years to get a robust longitudinal sample. Nevertheless, we acknowledge that some of our significant effects are underpowered and recognize the possibility that some nonsignificant findings (e.g., in relation to engineering majors at graduation) may have been significant with a larger sample size. Future work should seek to replicate this intervention with larger samples. Furthermore, because we did not ask mentors about their own engineering experiences, we were unable to test whether mentor success was related to mentee outcomes. For example, we did not measure whether mentors were successful in obtaining engineering internships, thus we could not test whether mentors who had

engineering internship experiences were more successful in guiding their mentees to procure internships. Future research should examine mentor outcomes and whether the beneficial effects of female mentorship remain after accounting for mentor success.

Future research should also compare the roles of belonging and confidence as mediating mechanisms over a long educational time course (note: results for belonging, anxiety, and intentions to pursue an engineering career are reported in the Supplementary Materials section). Our earlier article found that in the first year of college, female engineering students' feelings of belonging in engineering consistently mediated the effect of female peer mentors on career aspirations in engineering[32]. In contrast, our present investigation following the same female students for a longer period of time shows that when the entire college experience is analyzed, as in the current study, confidence in engineering abilities is a better mediator of the positive impacts of female peer mentors on female students' academic outcomes. These results suggest that the mechanism that predicts mentees' persistence and success in engineering may change over time during college. More research is needed to replicate these findings and explore the importance of different types of psychological mechanisms at different developmental stages of young people's academic careers.

Another difference emerged in student outcomes during the first year of college compared with a longer timeframe (across the entire college career plus one year post-graduation). In our previous article, we found that at the end of the first year, 100% of participants who had been assigned a female mentor remained in engineering majors, compared to 82% of participants assigned a male mentor and 89% of participants assigned no mentor[32]. Our present research shows that this difference was no longer significant at graduation. However, a closer examination of the data suggests that some participants who had female mentors switched out of engineering after their first year to a different STEM field—specifically computer science, mathematics, chemistry, or biology. Indeed, participants with female mentors were more likely to graduate with STEM degrees compared to participants in the other two conditions. Thus, it appears that having female mentors in engineering during their first year proved beneficial for female participants' persistence in STEM broadly, which is crucial given the undersupply of highly skilled talent across multiple STEM fields in the U.S. labor market and the high demand for employees with a variety of such skills.

It is also worth noting that female students' engineering and STEM grade point average (GPA) in college did not differ between mentor conditions ($p$s ≥ 0.457), nor did their GPA trajectories significantly change over time ($p$s ≥ 0.161) or differ by mentor condition ($p$s ≥ 0.504). Thus, the benefit of female mentorship on female participants' academic outcomes did not occur by improving mentees' grades. Instead, our findings suggest that improving their positive subjective experiences in engineering were more likely to be the causes of the observed benefits.

Some important future directions include replicating these findings across other STEM disciplines where females are also a minority (e.g., computer science, physics)[33] and examining mentoring interventions at later points in females' STEM careers (e.g., during graduate or postdoctoral traineeships, and early careers in STEM). Additionally, an important direction of future research is to investigate the generalizability of these findings to other identity groups that are underrepresented and negatively stereotyped in STEM (e.g., Black, Hispanic, Indigenous, and working-class college students who are first in their families to receive a Bachelor's degree)[1,2,55].

We propose that examining (and scaling up) research-driven solutions in academic and professional institutions to support the success of underrepresented students in STEM pathways and tracking the long-term impacts of these solutions are important steps to improving representation in STEM. Our findings indicate that a low-cost, light-touch mentoring relationship with a successful own-group peer during the transition to college yields dividends even after it has concluded, both for subjective indicators of confidence in overall engineering abilities, motivation in engineering courses, and emotional well-being as well as objective indicators of skills and persistence. Such a relational intervention, initiated in the transition to college, constitutes an important step towards equalizing the representation of tomorrow's scientists and engineers.

## Methods
### Participants
Our study participants (i.e., potential mentees) comprised of 150 female students majoring in engineering at the University of Massachusetts Amherst. We also recruited 58 student mentors (32 females, 26 males) based on faculty recommendations. Our study was approved by the University of Massachusetts Amherst Institutional Review Board. Data collection took place over 8 years, from 2011–2019. Participants were recruited during new student orientation for the 2011–12, 2012–13, 2013–14, and 2014–15 academic years. Most students were entering first-years (80%). The remainder were transfer students joining in their second year (20%). During their first year in the study, participants were randomly assigned to a female mentor ($n = 52$), male mentor ($n = 51$), or no mentor (control condition; $n = 47$). Participants were told that the study was examining factors related to success and development in engineering majors and were unaware that the study was related to mentoring. Participants consented to taking part in this long-term study and also gave consent for the researchers to access their college transcripts. Mentors were undergraduate students in their junior or senior year in the same engineering major as their mentee (e.g., electrical engineering, mechanical engineering) and gave consent to being a mentor.

At the time of the baseline survey, participants were 18.34 years of age, on average ($SD = 1.34$). Participants reported their sex and race based on National Science Foundation (NSF) demographic groupings at the time: participants were asked to identify whether they were male or female, and whether they identified as American Indian/Alaskan Native, Asian or Pacific Islander, Black (not of Hispanic origin), Hispanic, white (not of Hispanic origin), multiracial, or another ethnicity. All participants reported their sex as female. The sample was 67.3% white, 17.3% Asian, 5.3% multiracial, 2.7% Black, 2.7% Hispanic, and 2% other ethnicity. Participants were paid $20 for the baseline survey, $30 for the second survey, and $35 for the third survey during their first year. Participants were paid $10 for each follow-up survey in subsequent years. Peer mentors were paid $100 for each mentee they had. As some mentors had multiple mentees, we tested whether any of our effects could be attributed to individual mentors. We found that individual mentors did not contribute significant variance to any outcome variable ($p$s ≥ 0.251), thus, we did not control for this variable in our analyses.

### Procedure
During their first year at the university, participants completed a survey at 3 time-points during the academic year: (1) a baseline survey in August or September prior to meeting with their mentors, (2) a mid-year survey in January or February, and (3) an end-of-year survey in April or May. After the first year, when the mentoring relationship had officially ended, we asked participants to complete a follow-up survey each year during the spring semester (February to May) or summer, until they graduated with a Bachelor's degree plus one year post-graduation. The number of participants at each time point is listed in Table 2. On average, participants completed a total of 5.10 surveys; 69% of participants (104 out of 150) completed 5 or more surveys and 22% completed 4 surveys (33 out of 150). The number of surveys

**Table 2 | Number of participants across timepoints for continuous variables**

| | Year 1 (Baseline: Before Intervention) | Year 1 (Middle of Intervention) | Year 1 (End of Intervention) | Year 2 | Year 3 | Year 4 | Year 5+ |
|---|---|---|---|---|---|---|---|
| Number (percent) of survey respondents | 150 (100%) | 150 (100%) | 150 (100%) | 102 (68%) | 49 (33%) | 61 (41%) | 103 (69%) |
| Number (percent) of college transcripts collected | – | – | – | – | – | – | 150 (100%) |

Table 2 depicts the number and percentage of participants who completed the survey each year, as well as the percentage of college transcripts with major, courses taken, and grade information that were collected from the university registrar's office. Almost all participants completed their post-graduation survey during year 5 and up. A few participants (*n* = 6) who graduated during year 3 completed their post-graduation survey during year 4.

completed did not significantly differ by mentoring condition ($\chi^2(8, N = 150) = 7.06$, $p = 0.531$).

Peer mentors were primarily seniors, with some juniors, who were in good standing in their engineering major and played leadership roles in student-run professional clubs. Before being assigned a mentee, mentors attended a half-day training workshop at the beginning of the academic year. During the training, we emphasized two major points. First, mentors were asked to reflect on their first two years in engineering, identify any difficulties they had experienced, and the support or information that they wished they had received during that difficult time. After generating their personal experiences, mentors participated in a facilitated group discussion and collectively identified what experiences encouraged them and other students to persist in engineering and what other difficulties discouraged students and may lead them to drop out of engineering. The ideas generated by mentors were summarized in the form of a mentoring guide and provided to all mentors, together with topics of discussion to raise with their mentees (e.g., providing advice on academic coursework; providing tutoring help; helping mentees develop plans for college and careers including how to search for research assistantships and internships; providing social support; connecting mentees with other students and student clubs). Our mentoring guide can be found at https://osf.io/968ta/. Mentors were encouraged to share this information with their mentees and to be a source of support. Second, mentors were asked to meet with their mentees once a month throughout the academic year and engage with them on any topic in their mentoring guide based on mentee need. On average, mentors and mentees met four times for an hour each during mentees' first-year in college. Mentors were told that their primary goal was to be their mentee's friend and ease their transition into college. They were encouraged to meet over social activities. Finally, mentors were provided the names of their mentees and asked to reach out to their mentees to initiate contact. They were asked to complete a brief online survey after each meeting to keep a record of the meeting, including a summary of the topics discussed.

**Measures**

For our measure of confidence, participants completed two items that measured how confident they were about their overall ability in engineering, drawn from previous research[17,32]. These items were: "Do you think you have a talent for engineering?" and "How confident do you feel about your engineering ability?" They answered on scales from 1 (not at all) to 7 (very much). Responses were averaged to create an index of confidence in engineering ($\alpha$s between 0.78 to 0.92). These items were asked at three time-points during participants' first-year in college and subsequently once a year until one year post-graduation.

Five items were used to measure participants' self-reported experiences of motivation in the context of their engineering courses, which were drawn from prior research[29,56–59]. The five motivation items were: "I have the skills and abilities to be successful in my engineering-related classes this year;" "I will be able to overcome any difficulties I experience in my engineering-related classes this year;" "I have what it takes to deal with my engineering-related classes;" "I am prepared to deal with my engineering-related classes;" and "I feel

confident about my engineering related classes this year." If participants were responding to the survey after switching majors or after graduating from college, they answered the questions pertaining to the most recent engineering classes they had taken (e.g., "I had the skills and abilities needed to be successful in my engineering-related classes"). Participants reported how much they agreed with each statement on a scale from 1 (not at all true) to 7 (very true). Responses to these items were averaged to form an index of motivation ($\alpha$s between 0.79–0.95). These items were asked at three time-points during participants' first-year in college and subsequently once a year until one-year post-graduation.

At the end of the first year and in each subsequent survey in college, participants were asked to report whether they had an engineering internship in the past year. In the post-college survey, participants reported whether they had an engineering internship during their final year of college. Participants' responses were compiled to create a dichotomous index of whether they had any engineering internships while in college (0 = no engineering internship, 1 = had one or more engineering internships). We treated this as a dichotomous measure for two reasons. First, there was a good bit of variability in the number of years it took participants to graduate from college (between 3–5 years), which would alter the number of internships that they could have had in college, with more internship possibilities for students who took longer to graduate. Second, the number of participants who completed the survey varied from year to year, which meant that a non-response in a given year would count as zero internships for that year, which might not be accurate. For both these reasons, engineering internships was treated as a categorical variable.

Using participants' undergraduate transcripts, we created two dichotomous variables. First, we coded whether each participant earned an engineering degree (0 = no engineering degree, 1 = earned an engineering degree). We also coded whether participants graduated with a STEM degree (0 = no STEM degree, 1 = earned a STEM degree). This variable was created by using the U.S. Department of Education's definition of STEM[48] which includes biological sciences, physical sciences, computer sciences, engineering, mathematics and statistics, and science technologies. We did not include social and behavioral sciences in this definition of STEM because social and behavioral sciences tend to have more female students at the undergraduate level. In some social science majors, females are the majority at the undergraduate level[60].

In each survey, we asked participants how likely they were to pursue a Master's degree or Ph.D. in engineering on a 7-point scale from 1 (not at all) to 7 (very much)[17]. These items were asked at three time-points during participants' first-year in college and once a year in every subsequent year until one-year post-graduation.

During the last time-point of their mentorship year (i.e., spring of their first year), we asked participants to self-report their mental health over the past year ("In terms of your overall mental health, how psychologically healthy and happy did you feel during this past academic year?") on a scale of 1 (not at all) to 7 (very healthy and happy). This item was repeated each year until one year post-college graduation.

**Reporting summary**

Further information on research design is available in the Nature Portfolio Reporting Summary linked to this article.

## Data availability

The participant data and syntax for each model in this study have been deposited in the Open Science Framework (OSF) repository and can be found at https://osf.io/968ta/. Source data are provided with this paper.

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

## Acknowledgements

We are grateful to the women who participated in this multi-year experiment and to the mentors who dedicated their time. We also thank Drs. Paula Rees and Bernhard Schliemann of the College of Engineering for their help in recruiting participants and mentors. We thank Dr. Tara Dennehy for her role in training mentors and collecting data during the early years of this project and Dr. Holly Laws for her advice on growth modeling and multilevel mediation. Finally, we thank members of the Implicit Social Cognition Laboratory for their role in data collection, data entry, and coding, and for their suggestions on an earlier draft of this paper. This research was supported by National Science Foundation Grant GSE 1132651 to N.D. (PI).

## Author contributions

N.D. designed research and secured grant funding to support the work. K.C.T. collected the data. D.J.W. performed the data analysis and wrote the first draft. K.C.T. and N.D. reviewed and revised the paper.

## Competing interests

The authors declare no competing interests.
