## [Peer Review File · Nature Communications]

Female Peer Mentors Early in College Have Lasting Positive Impacts on Female Engineering Students that Persist Beyond GraduationREVIEWER COMMENTS

Reviewer #1 (Remarks to the Author):

Thank you for the opportunity to review “The Enduring Impact of Female Peer Mentors in the First Year of College on Women Students in Engineering from College Entry through Graduation”. This paper examines the effects of a field experiment in which female students in engineering receive mentoring from a male or female mentor or not mentor during their first-year. Results show that receiving mentoring, especially from a woman, had positive effects over the course of their major and in the one-year follow-up after graduation (on subjective experiences [e.g., well-being] and objective outcomes [e.g., obtaining a STEM degree]). The paper focuses on a highly societally relevant topic of how to retain women in engineering, a field in which women are strongly underrepresented. I was impressed by the rigorous intervention set up by the researchers and their ability to follow-up for several years. However, I did have several questions and concerns about the paper that I will outline below. These mainly relate to the contribution of this paper compared to their 2017 PNAS paper, the sample size, and some of their methodological and data analytical decisions. I hope that my suggestions will be helpful for the authors to further develop this interesting and relevant work.

(1) INTRODUCTION

My first concern relates to the addition of this paper compared to their 2017 PNAS paper on the same project (but with less follow-up data). The authors state that this paper is a substantial expansion compared to that paper by (1) examining whether the first-year mentorship impact endured once mentorship concluded, and (2) whether this durable impact emerged on objective indicators too. However, in their 2017 paper they did examine question (1) by testing the effects 1 year after mentoring concluded, and they examined question (2) by testing the impact on GPA and retention in year 2 (so one year after the mentoring concluded). I agree with the authors that it is interesting to examine the longer term impact on this sample during the entirety of their major, but I did feel that their phrasing of how this paper is an addition to their previous paper was not entirely accurate. It also took quite a while before the authors mention this 2017 PNAS paper (until the end of the introduction), while it did seem highly relevant in earlier sections too. I found this especially noteworthy when they discussed the role of the gender of the mentor and related previous work on page 6. The authors state that “...because these studies were conducted in short-term experiments using experimenters posing as role models it is unclear whether these findings will generalize to real-world academic settings.” However, in their 2017 PNAS paper they already established in a randomized controlled experiment that male mentors did not protect women’s outcomes over the course of 2 years. Hence, I was not convinced that this paper is a substantial contribution in that regard compared to their previous paper. In my opinion, the main potential contribution of this paper is whether the intervention had impact over the course of 4 years, compared to their previous work where they looked at the impact over the course of 2 years. Whether or not the intervention had an impact over the course of 4 years too and not ‘just’ 2 years is definitely an important question, especially since most published intervention work in this area only looks at short-term effects. However, I’m not sure it is a contribution significant enough for this journal, especially given that a large portion of the data is exactly the same as in the 2017 paper.

(2) METHODS

Given that the 4-year (vs. 2-year) contribution of this paper is its main contribution, it is important to note that I had concerns about the amount of data the analyses are based on. The abstract mentions a sample size of 150, but this seems to be the number of women in the first year of data collection or the total number of women completing at least one survey (this was not entirely clear from the Procedure)? It would be helpful to see a table with the number of participants per time point. How many participants actually completed all surveys? Because in the end that is your final sample size, is it not? The 2017 PNAS paper mentions that 78 women completed the time 4 survey (the one-year follow-up). This already leaves a relatively low sample size (considering there are 3 experimental conditions), so I wonder what the sample size is in the other follow-ups? I know that Mplus uses all data points available, but still the number of data points in the final surveys must be lower. Can the authors say something about their power? Furthermore, as a consequence of their analysis choice

the comparisons between surveys are not based on the same sample. This also leaves the possibility that a level on a variable at time point X+1 is higher/lower compared to time point X because you are looking at a different sample at time point X+1 and not because their level on the variable increased/reduced. Can the authors exclude this possibility?

The authors created a ratio of anxiety and motivation, based on the threat and challenge literature where this is common. This seemed to be a bit of a stretch to me, or at least the connection was not obvious to me. What do the findings show for anxiety and motivation as separate scales? Additionally, I wondered if self-efficacy would be a better label for their motivation scale, since the items measure the extent to which a person feels like they have the necessary skills.

(3) RESULTS

I had several concerns about the way the data was analyzed:

The analyses are always conducted examining the change in a variable compared to the baseline. This is an interesting way of looking at this data, but I did wonder what the difference between conditions was at each time point (so the level, not the change). Especially for the baseline survey I believe this is critical, because this would speak to the successfulness of randomization. Currently, the figures create the idea that there is no difference at baseline and that after that the conditions start to diverge, but I wondered if this is indeed correct?

I was a bit confused by the authors' use of Time in the models. The baseline was indicated as Time 0 and the others time points were scales in reference to the month in which it was obtained relative to baseline. So for example, if the baseline was assessed in September (0), then the mid-year survey in Year 1 in January would be (4), the one-year follow-up survey assessed in February in Year 2 would be (17), etc. – is that correct? It was unclear to me how dividing this time variable by 12 would then mean that the slopes indicate yearly change? In the figures it also seems that the mid-year and end-of-year surveys in Year 1 are combined into one point (Year 1), is that correct? How does that work?

The authors explained that they conducted multilevel models with the variables at level 1 and the mentor condition at level 2. However, the variables at level 1 are not just nested within condition but also within participants. Should that not be a level too? (so participants at level 2 and condition at level 3, with variables nested within participants and participants nested within conditions).

The mediation models are conducted with dichotomous dependent variables (whether or not they secured internships and obtained a STEM major). I am no expert on these types of analyses, but is this possible? Would a logistic regression not be more appropriate?

(4) DISCUSSION

The authors mention that “Our earlier article showed that during the first year of college, women’s feelings of belonging in engineering consistently mediated the effect of female peer mentors on career aspirations in engineering³⁴. In contrast, our present investigation shows that when the entire college experience is analyzed as a whole, lower anxiety relative to higher motivation mediated the effect of female peer mentors on women’s academic outcomes.” However, the authors only reported having tested anxiety relative to motivation as a mediator. Perhaps adding a sentence in the Results section referring to other mediation analyses in the supplemental materials that were not significant would be helpful here. More generally, I wondered why the authors did not report or test emotional well-being as a mediator? Protective well-being as a mechanism explaining why female mentors protect retention and other successes would make theoretically sense to me?

In the discussion the authors primarily focus on the parts of their findings showing that oftentimes participants in the no mentor or male mentor conditions showed a change (e.g., in decrease wellbeing) while participants in the female mentor condition showed no change – concluding that having a female mentor is more beneficial than a male (or no) mentor. This is indeed a very consistent finding, but I wonder if their conclusion is too strong as oftentimes they do not actually see a difference between the trajectories of male and female mentors (because the trajectory of the male mentor condition falls in between the other two conditions).

I found the authors' theorizing of different types of psychological mechanisms playing important roles at different developmental stages in students' careers (belonging in the beginning and anxiety/motivation towards the end) very interesting and convincing (but see my concerns above regarding the way anxiety/motivation was examined).

MINOR POINTS:

How many follow-up surveys were there? If the major is 3 years, that would mean that, compared to their 2017 paper (where they had 1 follow-up survey, so in year 2), there is 1 additional follow-up survey in year 3 and one post-graduation survey. Is that correct?

The 2017 PNAS paper also mentioned several scales that were not mentioned in this paper. Does this mean that these measures were not assessed in the additional follow-up surveys? If this is not the case, then it would be helpful to have a description of why these measures were not included in this paper and to have a list of measures in the supplemental materials (or a short overview of what results on these measures shows). For example, the measures Thoughts of switching majors and Intention to pursue engineering careers from their 2017 paper seem very relevant for this paper too?

The measures seemed face valid to me, but I did miss a discussion of what these measures were based on (which validated scales or previous studies?).

Reviewer #2 (Remarks to the Author):

Ms. No.: NCOMMS-21-21677

Title: The enduring impact of female peer mentors in the first year of college on women students in engineering from college entry through graduation

The authors report results from a longitudinal field study designed to experimentally examine the impact of peer mentors on the college experiences of women interested in engineering. This work builds upon their previous published work that examines outcomes only during the first year. By exploring the long-term repercussions of the one-year mentoring program, they reveal that having a female mentor promoted women's success in STEM, including more positive subjective experiences, STEM-related choices, and emotional well-being, more so than having no mentor or a male mentor.

Overall, there is much to like about this manuscript. It is very well-written and it offers a good overview of the state of the literature on psychological interventions to promote representation in STEM and the mentorship and role models literatures. Not only do the authors have clear expertise and command over these broad areas of research, this work aims to fill important lacunae. Most notably, there is a much greater need for controlled, experimental work on the role of mentors in promoting student success and flourishing. And critically, they offer a well-designed study not only testing mentorship relative to a control, but also examining the role of role model gender, to offer an important and nuanced test of social identity in the mentorship process. Additionally, while much of the intervention work focuses on grades, it is becoming apparent that honing in more specifically on motivation and interest is critical when the goal is increasing students in STEM, and this work does just that. Moreover, the long-term nature of this work is one of the biggest strengths. So much intervention work examines relatively short-term effects; following students through their undergraduate career offers us an important perspective on the enduring impact of mentors.

There are a number of ways this manuscript could be strengthened:

- On p. 6, the authors mention the importance of creating authentic relationships, something not often studied in the role model literature. Can they talk more about why authentic relationships matter, and in particular, in the context of the Stereotype Inoculation Model?

- Related to this, do they have any measures of the relationship between mentors/mentees? If so, are beneficial effects stronger for those who developed stronger relationships in year 1? Importantly, did they assess if people stayed in contact with their mentor beyond year 1? For example, it could be the case that the same-gender dyads were more likely to develop enduring relationships that help explain the findings. Or not, this is a complete conjecture. The point is, this would be good to examine. They do seem to have a measure of how often they met. Does this assessment moderate effects at all? I understand the sample size might make this difficult to look at.
- The sample size is a drawback to this research. The authors should offer power analyses and should discuss the many non-significant findings in light of the sample size and power.
- I believe the challenge/threat paradigm employed to examine anxiety and motivation as a ratio is an appropriate and meaningful way to look at these data. However, on p. 10 the argument “As in previous studies...” does not give the readers an understanding of why they would do this. They do discuss better on p. 23; they should bring their justification of their use of the ratio up earlier in the paper.
- Do the authors have information on whether the various mentors had already had, or were accepted to, an internship experience during that year when they were mentoring the students? It would strengthen the manuscript quite a bit to show that there were no differences in these experiences for the male and female mentors.
- The anxiety and motivation questions are all worded to be about engineering. How do the students who switched to another STEM major or a non-STEM major answering those questions over the follow-up years?
- The authors suggest on p. 17 that the process for the effectiveness of female and male mentors might differ. They should consider developing this more.
- Ideally this paper and these findings will play a role in promoting the development of mentorship programs in colleges and universities. To those ends, it would be great if the authors could provide more details, online, for the mentor training and guidance
- Small issue: Figures 4 and 5 are switched.

Overall, I am very positive about this manuscript. This is important work that fills a gap in our understanding of the power of mentors to promote women’s success in STEM.

Reviewer #3 (Remarks to the Author):

In this study, the authors report a longitudinal randomized controlled trial of an intervention designed to protect the subjective experiences and improve the participation of college women interested in engineering. Female students (N = 150) were randomly assigned to receive a female mentor, male mentor, or no mentor during their first year of college. Students reported on their experiences and education/career choices throughout college. The authors report that female peer mentors protected women’s subjective experiences and educational aspirations (relative to having no mentor or a male mentor) and that women with female peer mentors were more likely to engage in an internship (than those without a mentor) and were more likely to complete a major in STEM (than those with a male mentor). This research is tackling an interesting and important question, and this must have been a challenging study to conduct so I commend the researchers for their time and effort in doing so. These data are valuable, both from a practical and theoretical perspective. That being said, I do not think the manuscript is suitable for publication in its current form. My major concerns regard (1) the relation of this paper to the authors’ prior work and (2) the analytic approach used and the

presentation of results. I describe these and other concerns below. I hope that the authors find my comments helpful as they continue with this work.

Major Concern 1: Relation to Prior Work (Dennehy & Dasgupta, 2017)

1. Given that the initial impacts of this intervention have already been assessed, I thought it would be interesting to include a stronger theoretical background regarding how educational interventions influence outcomes over time. For example, what predicts whether interventions get less successful over time, remain steady in their effectiveness, or even increase in their effectiveness? Might any of those processes be at play here? The authors have already shown that this intervention “works”, but why does it continue to work over such a long period of time? Drawing from past literature on the effectiveness of education interventions over time might provide some useful context or ideas here.

2. I think more needs to be done to differentiate the authors’ prior paper and contributions from the current paper and contributions. The authors note that their earlier article addresses the impact of this program during the first year of college and that the current manuscript examines the long-term effects of first-year mentorship. This is misleading because the earlier article also examined outcomes after the first year of the program. For example, the current paper says that understanding long-term effects of first-year mentorship was absent from the earlier article and that this investigation seeks to understand “Did the impact of peer mentoring decay once mentorship concluded or did it endure?” But the earlier article reports that “The benefits of peer mentoring endured long after the intervention had ended.” I understand that the current paper examines outcomes even more distant in time from the initial intervention, but I think the authors need to be a little more careful in how they differentiate the two papers from each other. In addition, I think it would be useful to know that this paper is an extension of previously-published work earlier in the Introduction.

3. I noticed that the authors reported on other outcomes in their earlier article (e.g., intentions to pursue engineering careers). Did the authors assess these at the other timepoints reported on in the current manuscript as well? If so, it would be useful to know the results even if there is no effect of mentor condition on those outcomes.

4. I’m wondering why the authors chose to use different labels for the anxiety/motivation measure (which appears to be called threat and challenge in their initial paper) and the confidence measure (which appears to be called self-efficacy in their initial paper). Is there a specific reason for this? I think it would be easier for readers if consistent terms were used.

Major Concern 2: Analytic Approach and Presentation of Results

1. I noticed that there were 58 mentors and 150 mentees, which I assume means that some mentors mentored multiple students. However, it does not seem like the authors adjusted for potential nonindependence in outcomes between mentees of the same mentor. It certainly seems possible that mentees of the same mentor were more similar in their outcomes than mentees who had different mentors so accounting for this nonindependence, if it exists, would be important to do.

2. I had trouble understanding the analysis strategy (e.g., what all of the random effects were), and this might be because I am not familiar with multilevel modeling in MPlus. Given that not all readers will be familiar with MPlus, I suggest that the authors include equations to represent their models as these equations will be consistent across analysis software. One recommendation is to make sure that the equations presented are the ones used to generate the results. For example, it does not seem that the dummy codes presented were the ones used to test the A and B analyses (A: changes over time per condition and B: whether change in each condition is different from change in other conditions) listed in the first paragraph on page 10. If they were, then it was not clear how because the dummy codes seem to test the difference between having a female mentor and no mentor (first code) and then the difference between having a male mentor and no mentor (second code). I think presenting equations might be useful in clarifying exactly what analysis was done to produce what results, but other strategies may also work.

3. Dummy codes: It would be useful to note that the reference group is the no mentor condition.

4. Longitudinal outcomes:

a. I was expecting the authors to first report omnibus tests that showed whether the mentor condition significantly interacted with time to predict any of the outcomes as a way to control for Type I error rate. Then, I would expect them to do the A and B analysis strategies they listed on page 10. Is there a reason they did not conduct omnibus tests like this? Did they control for Type I error rate in a different way?

b. In addition to understanding changes over time, it would also be useful to know at what timepoints the outcomes differed between conditions. For example, was anxiety (relative to motivation) significantly lower for those with female mentors, relative to those with male mentors or no mentors, at the final timepoint of the study? Was it significantly different at any other timepoints as well? Knowing the answers to these questions would be useful in understanding how long it takes for this mentor intervention to yield effects.

c. Did the authors consider any nonlinear changes over time? Some interventions show significant effects early on which remain steady over time but do not increase, which would result in a nonlinear trajectory. Did the authors conduct any analyses to examine whether nonlinear trajectories might be more appropriate?

5. Figures: Can the authors include standard error or confidence interval bands in the figures? It would be interesting to know more about the variability associated with outcomes for people with female vs. male mentors. It might speak to whether some male mentors are as helpful as female mentors. If this is the case, it might ultimately be worthwhile to try to understand what factors allow male mentors to be helpful as well—given that women in engineering are likely to encounter more men at “mentor” levels than women at these levels.

Minor Methodological Questions

1. I'm curious about the distribution of the internship measure and why the authors decided to make this a binary measure. Were there many students who had more than one internship? If so, might there be value in using this as a “true” count measure, where all internships are counted?

2. I found it a little difficult to understand what percentage of students completed measures at each timepoint of the study. I suggest the authors include a table or figure that shows what percentage of students across each mentor condition participated at each timepoint.

3. Related to the prior point, are the data from the survey after graduation included here? The figures go up to year 4, so it seems like they aren't. If not, why not? Were the measures the same?

Minor Questions about the Discussion

1. I was surprised the authors did not address reasons for why the mentor intervention is effective, other than stereotype inoculation. For example, is it possible that female mentors behave in different ways? Is it possible they provide useful information or a perspective that male mentors don't?

2. Did belongingness not mediate the relationship between mentorship and academic outcomes here? The discussion (pages 17 to 18) seems to imply that it did not but I was not sure where that was reported.

3. I would be careful about saying that assignment to a male mentor or no mentor “increased” women's anxiety (bottom of p. 16). It's more likely that the “baseline” is an increase in anxiety over the course of college, and female mentors simply protect against the occurrence of this, rather than male

mentors or no mentors causing greater anxiety.

Reviewer Comments and Author Responses for NCOMMS-21-21677A

April 29, 2022

Reviewer #1 (Remarks to the Author):

Thank you for the opportunity to review “The Enduring Impact of Female Peer Mentors in the First Year of College on Women Students in Engineering from College Entry through Graduation”. This paper examines the effects of a field experiment in which female students in engineering receive mentoring from a male or female mentor or not mentor during their first-year. Results show that receiving mentoring, especially from a woman, had positive effects over the course of their major and in the one-year follow-up after graduation (on subjective experiences [e.g., well-being] and objective outcomes [e.g., obtaining a STEM degree]). The paper focuses on a highly societally relevant topic of how to retain women in engineering, a field in which women are strongly underrepresented. I was impressed by the rigorous intervention set up by the researchers and their ability to follow-up for several years. However, I did have several questions and concerns about the paper that I will outline below. These mainly relate to the contribution of this paper compared to their 2017 PNAS paper, the sample size, and some of their methodological and data analytical decisions. I hope that my suggestions will be helpful for the authors to further develop this interesting and relevant work.

(1) INTRODUCTION

My first concern relates to the addition of this paper compared to their 2017 PNAS paper on the same project (but with less follow-up data). The authors state that this paper is a substantial expansion compared to that paper by (1) examining whether the first-year mentorship impact endured once mentorship concluded, and (2) whether this durable impact emerged on objective indicators too. However, in their 2017 paper they did examine question (1) by testing the effects 1 year after mentoring concluded, and they examined question (2) by testing the impact on GPA and retention in year 2 (so one year after the mentoring concluded). I agree with the authors that it is interesting to examine the longer term impact on this sample during the entirety of their major, but I did feel that their phrasing of how this paper is an addition to their previous paper was not entirely accurate. It also took quite a while before the authors mention this 2017 PNAS paper (until the end of the introduction), while it did seem highly relevant in earlier sections too. I found this especially noteworthy when they discussed the role of the gender of the mentor and related previous work on page 6. The authors state that “...because these studies were conducted in short-term experiments using experimenters posing as role models it is unclear whether these findings will generalize to real-world academic settings.” However, in their 2017 PNAS paper they already established in a randomized controlled experiment that male mentors did not protect women’s outcomes over the course of 2 years. Hence, I was not convinced that this paper is a substantial contribution in that regard compared to their previous paper. In my opinion, the main potential contribution of this paper is whether the intervention had impact over the course of 4 years, compared to their previous work where they looked at the impact over the course of 2 years. Whether or not the intervention had an impact over the course of 4 years too and not ‘just’ 2 years is definitely an important question, especially since most published intervention work in this area only looks at short-term effects. However, I’m not sure it is a contribution significant enough for this journal, especially given that a large portion of the data is exactly the same as in

the 2017 paper.

We thank the reviewer for raising this issue. We have revised the manuscript to now raise the 2017 PNAS paper in the Abstract (p. 2) and early in the Introduction (pp. 6-7). We also clarify that in the 2017 paper, the 1-year follow-up after the intervention only included half the sample (52%, $n = 78$), since only that portion of our sample had completed their second year of college at the time of data analysis and publication. Our current manuscript is a substantial extension in two ways. First, in our current manuscript, we followed the full sample of students for 3-5 years in college through college graduation plus additionally one-year post-graduation. As such, we are now in a stronger position to speak about the enduring benefits of a one-year peer mentoring intervention in the first year of college on students' psychological, behavioral, and retention outcomes in STEM for the *entirety of their college life (3-5 years) plus 1 year past college graduation*. Our 2017 PNAS paper did not have the data to extensively test longitudinal hypotheses like we do here.

Second, in addition to the original set of variables reported in the 2017 paper (confidence, anxiety, motivation, belonging, major retention rates, and graduate school intentions), the current manuscript also includes new variables: 1) successful completion of engineering internships; and 2) emotional well-being in college. Furthermore, we provide a better accounting of major choice in this new paper compared to the prior one. Specifically, in the 2017 paper, retention in engineering majors was measured at the end of the first year of college, well before students' deadline for making final decisions about academic majors. At American universities, students must make this decision at the end of the second year of college and students often switch majors in the summer after the first year or during their second year of college. By measuring students' majors at college graduation, the current manuscript provides a more accurate picture of how first-year peer mentorship impacts students' final decisions about their academic majors, which has lasting impacts on their post-graduation careers. For these two reasons (extensive follow-up 3-5 years post-intervention and the inclusion of new measures), we believe that our current paper is a significant extension above and beyond the first paper.

(2) METHODS

Given that the 4-year (vs. 2-year) contribution of this paper is its main contribution, it is important to note that I had concerns about the amount of data the analyses are based on. The abstract mentions a sample size of 150, but this seems to be the number of women in the first year of data collection or the total number of women completing at least one survey (this was not entirely clear from the Procedure)? It would be helpful to see a table with the number of participants per time point. How many participants actually completed all surveys? Because in the end that is your final sample size, is it not? The 2017 PNAS paper mentions that 78 women completed the time 4 survey (the one-year follow-up). This already leaves a relatively low sample size (considering there are 3 experimental conditions), so I wonder what the sample size is in the other follow-ups? I know that Mplus uses all data points available, but still the number of data points in the final surveys must be lower.

In the revised paper, we provide a data table (see Table 1 on p. 38; also inserted below) that shows the number and percentage of participants who completed a survey at each time point as well as our percentage of participants for whom we had data for their major at graduation.

Because we utilized full information maximum likelihood (FIML) estimation, all data points were used in our final sample size (as opposed to listwise deletion, which would reduce the number of data points in the study to include only data from participants who completed measures at all time points). FIML estimation is the preferred method to handle missing data because simulation studies show that FIML estimation more accurately reflects the complete dataset than mean imputation and pairwise or listwise deletion (Enders, 2001). After accounting for participants' varying graduation dates (some graduated in 3 years; others in 4 or 5 years), our current data show that we have 74% complete data points. Recent simulation studies show that FIML estimates are stable and do not show signs of bias when there is 25% missing data, as we have here (Lee & Shi, 2021).

	Year 1 (Baseline: Before Intervention)	Year 1 (Middle of Intervention)	Year 1 (End of Intervention)	Year 2	Year 3	Year 4	Year 5+
Number (percent) of survey respondents	150 (100%)	150 (100%)	150 (100%)	102 (68%)	49 (33%)	61 (41%)	103 (69%)
Number (percent) of transcripts collected	-	-	-	-	-	-	150 (100%)

Can the authors say something about their power? Furthermore, as a consequence of their analysis choice the comparisons between surveys are not based on the same sample. This also leaves the possibility that a level on a variable at time point X+1 is higher/lower compared to time point X because you are looking at a different sample at time point X+1 and not because their level on the variable increased/reduced. Can the authors exclude this possibility?

Power analyses are now reported (see pp. 12-16). In response to Reviewer 1's question, because we analyzed our data using linear growth curve analyses with full information maximum likelihood (FIML), we estimate individual participants' slopes based on their available data points rather than estimating their specific values at each timepoint; this captures participants' change over time. These slopes are then aggregated and compared between mentor conditions.

The authors created a ratio of anxiety and motivation, based on the threat and challenge literature where this is common. This seemed to be a bit of a stretch to me, or at least the connection was not obvious to me. What do the findings show for anxiety and motivation as separate scales? Additionally, I wondered if self-efficacy would be a better label for their motivation scale, since the items measure the extent to which a person feels like they have the necessary skills.

Based on Reviewer 1's advice, we now analyze and report anxiety and motivation as separate variables in our results (anxiety is reported in the Supplementary Materials). Our measure of confidence is actually self-efficacy (as reported in Dennehy & Dasgupta, 2017). We chose to use

the term confidence because it is more easily understandable to a non-psychologist audience whereas self-efficacy is a disciplinary jargon. We now include a footnote on p. 26 to specify this change in terms.

(3) RESULTS

I had several concerns about the way the data was analyzed:

The analyses are always conducted examining the change in a variable compared to the baseline. This is an interesting way of looking at this data, but I did wonder what the difference between conditions was at each time point (so the level, not the change). Especially for the baseline survey I believe this is critical, because this would speak to the successfulness of randomization. Currently, the figures create the idea that there is no difference at baseline and that after that the conditions start to diverge, but I wondered if this is indeed correct?

We found no baseline differences between conditions in participants' feelings of belonging, confidence, motivation, socioemotional health, graduate degree intentions, and engineering career interest. Only one statistical difference emerged at baseline in anxiety between the male mentor condition vs. control such that participants with male mentors showed *less* anxiety than controls at baseline. There was no difference in anxiety between male vs. female mentor condition. All this is now reported in Table S2 in the Supplementary Materials (p. 4) document. Given that all but one of our variables had no baseline differences, this shows the success of random assignment to condition. The figures are shown without baseline levels for ease of reading because our research questions focused on change over time rather than mean levels.

I was a bit confused by the authors' use of Time in the models. The baseline was indicated as Time 0 and the others time points were scales in reference to the month in which it was obtained relative to baseline. So for example, if the baseline was assessed in September (0), then the mid-year survey in Year 1 in January would be (4), the one-year follow-up survey assessed in February in Year 2 would be (17), etc. – is that correct? It was unclear to me how dividing this time variable by 12 would then mean that the slopes indicate yearly change? In the figures it also seems that the mid-year and end-of-year surveys in Year 1 are combined into one point (Year 1), is that correct? How does that work?

The three first-year surveys are *not* combined into one timepoint. Each survey has their own specific timepoint. Besides that, Reviewer 3's conception of time is correct such that a mid-year survey in January would be marked as 4 months after the baseline survey in September of the student's Year 1 (Time 0). A one-year follow-up in February of Year 2 would be 17 months after Time 0. These values are then divided by 12 for the ease of interpreting effects (i.e., a regression estimate would refer to the change across 12 months or one year).

Since there was variability in when participants completed each follow-up survey (i.e., from winter to late summer), we elected to use the exact month that their survey was taken relative to Time 0, rather than aggregate all surveys for each year without regard to the passage of time (month). In our graphs, we designate yearly timepoints for ease of understanding. But the data analysis used the real timepoints (specific month since Time 0) to estimate slopes for each participant and condition.

The authors explained that they conducted multilevel models with the variables at level 1 and the mentor condition at level 2. However, the variables at level 1 are not just nested within condition but also within participants. Should that not be a level too? (so participants at level 2 and condition at level 3, with variables nested within participants and participants nested within conditions).

Our data is best suited for a two-level model. Level 1 is within participants (variables that change over time within participant), while Level 2 contains variables that are between participants (treatment condition). If condition were relegated to Level 3 and treated as a cluster variable (similar to participant ID at Level 2), we would be unable to observe condition differences. We conducted the same type of multilevel models that were reported in our earlier investigation (Dennehy & Dasgupta, 2017).

The mediation models are conducted with dichotomous dependent variables (whether or not they secured internships and obtained a STEM major). I am no expert on these types of analyses, but is this possible? Would a logistic regression not be more appropriate?

In response to Reviewer 1's suggestion, we have now updated mediation models for dichotomous variables utilizing logistic regressions.

(4) DISCUSSION

The authors mention that “Our earlier article showed that during the first year of college, women’s feelings of belonging in engineering consistently mediated the effect of female peer mentors on career aspirations in engineering³⁴. In contrast, our present investigation shows that when the entire college experience is analyzed as a whole, lower anxiety relative to higher motivation mediated the effect of female peer mentors on women’s academic outcomes.” However, the authors only reported having tested anxiety relative to motivation as a mediator. Perhaps adding a sentence in the Results section referring to other mediation analyses in the supplemental materials that were not significant would be helpful here. More generally, I wondered why the authors did not report or test emotional well-being as a mediator? Protective well-being as a mechanism explaining why female mentors protect retention and other successes would make theoretically sense to me?

Emotional well-being did not mediate the female mentor effect on academic outcomes. These analyses are now reported on pp. 17-18. We also include motivation as a mediator on its own, rather than the anxiety relative to motivation ratio.

In the discussion the authors primarily focus on the parts of their findings showing that oftentimes participants in the no mentor or male mentor conditions showed a change (e.g., in decrease wellbeing) while participants in the female mentor condition showed no change – concluding that having a female mentor is more beneficial than a male (or no) mentor. This is indeed a very consistent finding, but I wonder if their conclusion is too strong as oftentimes they do not actually see a difference between the trajectories of male and female mentors (because the trajectory of the male mentor condition falls in between the other two conditions).

In our updated analyses that includes more data points for each participant extending one-year post-graduation, we now find significant differences between the trajectories of participants in the male vs. female mentor conditions for confidence, motivation, and emotional well-being. Based on this, we believe that there is consistent evidence that a female mentor is more beneficial for women in comparison to male mentors.

I found the authors' theorizing of different types of psychological mechanisms playing important roles at different developmental stages in students' careers (belonging in the beginning and anxiety/motivation towards the end) very interesting and convincing (but see my concerns above regarding the way anxiety/motivation was examined).

Thank you. As noted above, we now examine anxiety and motivation separately.

MINOR POINTS:

How many follow-up surveys were there? If the major is 3 years, that would mean that, compared to their 2017 paper (where they had 1 follow-up survey, so in year 2), there is 1 additional follow-up survey in year 3 and one post-graduation survey. Is that correct?

Reviewer 1's understanding is correct. After the initial intervention year, participants were asked to complete one survey each year they were in college, plus a survey 1 year after college graduation. If a participant graduated in 4 years, they would have completed 4 follow-up surveys after the first intervention year (in their second, third, and fourth year of college, and one year post-graduation). We now include a table with the number of surveys we collected at each time point on p. 38 of the revised manuscript.

The 2017 PNAS paper also mentioned several scales that were not mentioned in this paper. Does this mean that these measures were not assessed in the additional follow-up surveys? If this is not the case, then it would be helpful to have a description of why these measures were not included in this paper and to have a list of measures in the supplemental materials (or a short overview of what results on these measures shows). For example, the measures Thoughts of switching majors and Intention to pursue engineering careers from their 2017 paper seem very relevant for this paper too?

We did not test for thoughts of switching majors after the second year because students are not able to switch majors after their second year of college. We now report analyses for intentions to pursue engineering careers on p. 1 of the Supplementary Materials.

The measures seemed face valid to me, but I did miss a discussion of what these measures were based on (which validated scales or previous studies?).

We now include citations for all measures on pp. 26-28.

Reviewer #2 (Remarks to the Author):

Ms. No.: NCOMMS-21-21677

Title: The enduring impact of female peer mentors in the first year of college on women students in engineering from college entry through graduation

The authors report results from a longitudinal field study designed to experimentally examine the impact of peer mentors on the college experiences of women interested in engineering. This work builds upon their previous published work that examines outcomes only during the first year. By exploring the long-term repercussions of the one-year mentoring program, they reveal that having a female mentor promoted women's success in STEM, including more positive subjective experiences, STEM-related choices, and emotional well-being, more so than having no mentor or a male mentor.

Overall, there is much to like about this manuscript. It is very well-written and it offers a good overview of the state of the literature on psychological interventions to promote representation in STEM and the mentorship and role models literatures. Not only do the authors have clear expertise and command over these broad areas of research, this work aims to fill important lacunae. Most notably, there is a much greater need for controlled, experimental work on the role of mentors in promoting student success and flourishing. And critically, they offer a well-designed study not only testing mentorship relative to a control, but also examining the role of role model gender, to offer an important and nuanced test of social identity in the mentorship process. Additionally, while much of the intervention work focuses on grades, it is becoming apparent that honing in more specifically on motivation and interest is critical when the goal is increasing students in STEM, and this work does just that. Moreover, the long-term nature of this work is one of the biggest strengths. So much intervention work examines relatively short-term effects; following students through their undergraduate career offers us an important perspective on the enduring impact of mentors.

There are a number of ways this manuscript could be strengthened:

- On p. 6, the authors mention the importance of creating authentic relationships, something not often studied in the role model literature. Can they talk more about why authentic relationships matter, and in particular, in the context of the Stereotype Inoculation Model?

To address this question, we have now added additional literature review and discussion in the role model paragraph on p. 5. We argue that authentic long-term relationships with role models and mentors differ substantially from brief exposure to, or fleeting interactions with, short-term role models. Authentic long-term relationships allow mentees to seek advice, get assistance when they struggle, expand their academic and professional networks through their role model or mentor's contacts, and get socialized by their role model over time. Consistent with our predictions, past research demonstrates that authentic relationships with faculty mentors in STEM enhances student outcomes. For example, research shows that engineering students who have authentic relationships with faculty members who give personalized attention and mentoring, are more likely to persist and perform well in their classes (Vogt, 2008). Black students at historically Black universities who work directly with faculty when they struggle, are

more likely to thrive (Gasman & Nguyen, 2019) as compared to students in predominantly white universities where there is less attention to fostering one-on-one relationships between students and faculty (Davis & Finelli, 2007). In the context of the Stereotype Inoculation Model, we predict that opportunities to form authentic relationships with role models who share one's marginalized identity will have particularly important benefits for mentees by reinforcing their confidence, belonging, and abilities over time. Although past research on fleeting role model interventions have shown increased students' interest in STEM and improved performance on a brief math task or a one-time course, little is known about whether and when these benefits extend to long-term real-life outcomes (e.g., decisions about academic majors, work experience, post-college experiences).

- Related to this, do they have any measures of the relationship between mentors/mentees? If so, are beneficial effects stronger for those who developed stronger relationships in year 1? Importantly, did they assess if people stayed in contact with their mentor beyond year 1? For example, it could be the case that the same-gender dyads were more likely to develop enduring relationships that help explain the findings. Or not, this is a complete conjecture. The point is, this would be good to examine. They do seem to have a measure of how often they met. Does this assessment moderate effects at all? I understand the sample size might make this difficult to look at.

As reported in our original manuscript (p. 3 of the Dennehy & Dasgupta, 2017 Supplementary Materials), mentees were asked how closely they identified with their mentor (e.g., how similar they felt to their peer mentors, how much they felt personally connected with mentors, or whether they could imagine themselves achieving a similar level of success as their peer mentor). Identification with their mentor did not differ by mentor gender. When conducting correlations between this identification variable and our outcomes, we find that those who reported identifying more with their mentors showed a corresponding increase in confidence, motivation, intentions to pursue graduate school in engineering, and intentions to pursue an engineering career ($r_s > .20, p_s < .045$). Mentees who identified more with their mentors were also more likely to secure and successfully complete engineering internships, and graduate with engineering majors and STEM majors ($r_s > .20, p_s < .038$). However, the magnitude of all these correlations did not vary by mentor gender.

In terms of frequency of mentoring meetings, the number of meetings was positively correlated with an increase in belonging and a decrease in threat ($|r_s| > .22, p_s < .027$). These correlations did not differ by condition. When separated by condition, those in the female mentor condition who met more often with their mentors were more likely to graduate with a STEM degree ($r = .32, p = .032$). For those in the male mentor condition, number of meetings was not correlated with graduating with a STEM degree ($r = .07, p = .678$). However, the interaction effect (frequency of meeting x mentor condition) on STEM degrees did not reach significance ($p = .090$).

Given that these correlations did not vary by mentor gender, we did not include them in the current manuscript. However, we are happy to include them in the Supplementary Materials or discuss them in the manuscript if requested.

Additionally, we did not assess if people stayed in contact with their mentor beyond year 1, but we will keep this suggestion in mind for any future studies.

- The sample size is a drawback to this research. The authors should offer power analyses and should discuss the many non-significant findings in light of the sample size and power.

Power analyses are now reported (see pp. 12-16). We also include a discussion of limitations related to sample size on pp. 20-21.

- I believe the challenge/threat paradigm employed to examine anxiety and motivation as a ratio is an appropriate and meaningful way to look at these data. However, on p. 10 the argument “As in previous studies...” does not give the readers an understanding of why they would do this. They do discuss better on p. 23; they should bring their justification of their use of the ratio up earlier in the paper.

We now separate our anxiety/motivation ratio analyses and report them separately.

- Do the authors have information on whether the various mentors had already had, or were accepted to, an internship experience during that year when they were mentoring the students? It would strengthen the manuscript quite a bit to show that there were no differences in these experiences for the male and female mentors.

Unfortunately, we did not collect these data from our mentors, but in hindsight, we wish we had. We thank the review for this idea and plan to keep it in mind for future studies.

- The anxiety and motivation questions are all worded to be about engineering. How do the students who switched to another STEM major or a non-STEM major answering those questions over the follow-up years?

We have now clarified on pp. 26-27 that participants who switched majors were asked to answer these questions based on their most recent experiences in engineering. At each survey time point, participants who had switched majors were asked to reflect back on how they felt the last time they took engineering classes specifically for the motivation, anxiety, and belonging measures. We chose this approach because it allowed us to retain these participants in our dataset and capture their experiences (because their attrition from engineering and the reasons for it are part of the phenomenon we're interested in) instead of treating them as missing data. We recognize that while these students were reflecting back on their recent past, other participants who stayed in engineering were reflecting on their current experiences.

- The authors suggest on p. 17 that the process for the effectiveness of female and male mentors might differ. They should consider developing this more.

When reanalyzing our data using expanded data points for participants 1-year past graduation (rather than just through graduation), we no longer see benefits of male mentors across time. So, we have removed the language on p. 17 from our manuscript.

- Ideally this paper and these findings will play a role in promoting the development of mentorship programs in colleges and universities. To those ends, it would be great if the authors could provide more details, online, for the mentor training and guidance

Thanks for the suggestion. We have posted the mentorship program process and training materials in the Open Science Framework (OSF) website, and it is open access to all. The link to this folder is provided on p. 29 of the manuscript.

- Small issue: Figures 4 and 5 are switched.

This has been corrected.

Overall, I am very positive about this manuscript. This is important work that fills a gap in our understanding of the power of mentors to promote women's success in STEM.

Thank you.

Reviewer #3 (Remarks to the Author):

In this study, the authors report a longitudinal randomized controlled trial of an intervention designed to protect the subjective experiences and improve the participation of college women interested in engineering. Female students (N = 150) were randomly assigned to receive a female mentor, male mentor, or no mentor during their first year of college. Students reported on their experiences and education/career choices throughout college. The authors report that female peer mentors protected women's subjective experiences and educational aspirations (relative to having no mentor or a male mentor) and that women with female peer mentors were more likely to engage in an internship (than those without a mentor) and were more likely to complete a major in STEM (than those with a male mentor). This research is tackling an interesting and important question, and this must have been a challenging study to conduct so I commend the researchers for their time and effort in doing so.

These data are valuable, both from a practical and theoretical perspective. That being said, I do not think the manuscript is suitable for publication in its current form. My major concerns regard (1) the relation of this paper to the authors' prior work and (2) the analytic approach used and the presentation of results. I describe these and other concerns below. I hope that the authors find my comments helpful as they continue with this work.

Major Concern 1: Relation to Prior Work (Dennehy & Dasgupta, 2017)

1. Given that the initial impacts of this intervention have already been assessed, I thought it would be interesting to include a stronger theoretical background regarding how educational interventions influence outcomes over time. For example, what predicts whether interventions get less successful over time, remain steady in their effectiveness, or even increase in their effectiveness? Might any of those processes be at play here? The authors have already shown that this intervention "works", but why does it continue to work over such a long period of time? Drawing from past literature on the effectiveness of education interventions over time might provide some useful context or ideas here.

We thank Reviewer 3 for this important suggestion. We offer two theory-driven mechanisms to explain why female mentors were more effective than male mentors (now included in the Discussion on pp. 19-20). First, sociological research shows that for students who are minorities in a given field, developing a strong social network of people who share their marginalized identity in the target profession is especially helpful for professional success. In one study, Yang et al. (2019) found that female students in MBA programs were more successful in getting high-ranked jobs after graduation if they had a close network of women who advised them about where to apply, how to interview, and offered other background information about companies. Advice from close female connections (vs. male connections) may be especially effective because the women giving informal advice are more cognizant of potential barriers about which male peers may not be aware. In general, access to high-value social networks gives people an advantage in finding employment; people who are minoritized in a profession often have less access to professional connections that open doors (Pedulla & Pager, 2019; Trimble & Kmec, 2011). By connecting participants with female mentors in the first year of college, we may have provided new students their first connection in engineering. They may have used this relationship to expand their social network in engineering across the 3-5 years of college. The connections

made through the female mentors (as compared to the male mentors) may have been more valuable for women mentees' success, similar to Yang et al. (2019). Our finding that women with female mentors secured more engineering internships in college suggests that perhaps female mentors connected their mentees to recruiters and hiring managers with whom they had relationships. Future research should directly test if same-sex mentoring relationships expands mentees' access to high-value social networks.

A second potential mechanism may be that female mentors elicited a change in their mentees' mindset in a critical transitional period of their life. As theorized by Walton (2014), interventions may have a long-term impact if they change "recursive processes" early on (p. 80). If an intervention is able to alter individuals' way of thinking and help them gain more constructive mindsets regarding their learning in the beginning of a new environment, they will be more likely to persist in their education. In contrast, if students remain caught in cycles of psychological threat and poor performance, they will be less likely to persist. Applying mindset change to our research, the benefit of female (as compared to male) mentors may have persisted because women students were able to develop constructive mindsets regarding engineering and develop their professional identity as a future engineer through support from same-sex mentors and by envisioning those mentors as their future self. Future research should directly examine these and other potential mechanisms driving the success of same-sex mentorship interventions that have long-lasting effects.

2. I think more needs to be done to differentiate the authors' prior paper and contributions from the current paper and contributions. The authors note that their earlier article addresses the impact of this program during the first year of college and that the current manuscript examines the long-term effects of first-year mentorship. This is misleading because the earlier article also examined outcomes after the first year of the program. For example, the current paper says that understanding long-term effects of first-year mentorship was absent from the earlier article and that this investigation seeks to understand "Did the impact of peer mentoring decay once mentorship concluded or did it endure?" But the earlier article reports that "The benefits of peer mentoring endured long after the intervention had ended." I understand that the current paper examines outcomes even more distant in time from the initial intervention, but I think the authors need to be a little more careful in how they differentiate the two papers from each other. In addition, I think it would be useful to know that this paper is an extension of previously-published work earlier in the Introduction.

We have revised the manuscript to now raise the 2017 PNAS paper in the Abstract (p. 2) and early in the Introduction (pp. 6-7). We also clarify that in the 2017 paper, the 1-year follow-up after the intervention only included half the sample (52%, $n = 78$), since only that portion of our sample had completed their second year of college at the time of data analysis and publication. Our current manuscript is a substantial extension in two ways. First, in our current manuscript, we followed the full sample of students for 3-5 years in college through college graduation plus additionally one-year post-graduation. As such, we are now in a stronger position to speak about the enduring benefits of a one-year peer mentoring intervention in the first year of college on students' psychological, behavioral, and retention outcomes in STEM for the *entirety of their college life (3-5 years) plus 1 year past college graduation*. Our 2017 PNAS paper did not have the data to extensively test longitudinal hypotheses like we do here.

Second, in addition to the original set of variables reported in the 2017 paper (confidence, anxiety, motivation, belonging, major retention rates, and graduate school intentions), the current manuscript also includes new variables: 1) successful completion of engineering internships; and 2) emotional well-being in college. Furthermore, we provide a better accounting of major choice in this new paper compared to the prior one. Specifically, in the 2017 paper, retention in engineering majors was measured at the end of the first year of college, well before students' deadline for making final decisions about academic majors. At American universities, students must make this decision at the end of the second year of college and students often switch majors in the summer after the first year or during their second year of college. By measuring students' majors at college graduation, the current manuscript provides a more accurate picture of how first-year peer mentorship impacts students' final decisions about their academic majors, which has lasting impacts on their post-graduation careers. For these two reasons (extensive follow-up 3-5 years post-intervention and the inclusion of new measures), we believe that our current paper is a significant extension above and beyond the first paper.

3. I noticed that the authors reported on other outcomes in their earlier article (e.g., intentions to pursue engineering careers). Did the authors assess these at the other timepoints reported on in the current manuscript as well? If so, it would be useful to know the results even if there is no effect of mentor condition on those outcomes.

We now report analyses for intentions to pursue engineering careers on p. 1 of the Supplementary Materials. We find that women in the no mentor condition significantly declined in their intentions over time, while women in the female and male mentor conditions' intentions remained steady. However, given that these slopes did not differ between conditions, we are cautious to interpret this as evidence for a mentor benefit on this outcome.

4. I'm wondering why the authors chose to use different labels for the anxiety/motivation measure (which appears to be called threat and challenge in their initial paper) and the confidence measure (which appears to be called self-efficacy in their initial paper). Is there a specific reason for this? I think it would be easier for readers if consistent terms were used.

Anxiety captures an avoidance orientation in relation to engineering whereas motivation captures an approach orientation toward engineering. In the social psychological literature, these variables are typically called threat (anxiety) and challenge (motivation) (e.g., Vick et al., 2008). Because the readership of *Nature Communications* is broad and unlikely to be social psychologists, we replaced the threat vs. challenge jargon, which might be misunderstood given these terms do not align with everyday English meaning of threat and challenge. We chose to go with more user-friendly everyday language (anxiety vs. motivation) to suit a wider audience. Our measure of confidence is the same as self-efficacy. Again, we chose to use the term confidence because it is more easily understandable to a non-psychologist audience whereas self-efficacy is a disciplinary jargon. We now include a footnote on p. 26 to specify this change in terms.

Major Concern 2: Analytic Approach and Presentation of Results

1. I noticed that there were 58 mentors and 150 mentees, which I assume means that some mentors mentored multiple students. However, it does not seem like the authors adjusted for

potential nonindependence in outcomes between mentees of the same mentor. It certainly seems possible that mentees of the same mentor were more similar in their outcomes than mentees who had different mentors so accounting for this nonindependence, if it exists, would be important to do.

Following Reviewer 3's suggestion, we conducted three-level unconditional models (level 1: within participants capturing variations across time, level 2: individual differences between participants, level 3: mentor cluster ID) for each continuous variable, to assess intraclass correlations and variance for the mentor cluster variable. Mentor cluster was the label we gave the variable that captures mentees nested within a common mentor. An intraclass correlation is the ratio of the variance in a given level (in this case, the mentor cluster variable) relative to the total variance in participants' responses. The intraclass correlations for our mentor cluster variable were small (ICCs < 0.08) and the variances were all nonsignificant ($ps > .25$; see table below). This means very little variance in participants' responses on each dependent variable can be attributable to having a common mentor. In other words, nonindependence among mentees who worked with the same mentor is not a problem in our data. Furthermore, due to this limited variance, three-level conditional models that included mentor condition (male/female mentors) plus the mentor cluster variable were unable to converge. Given these additional analyses, we believe that there is little chance that our results are due to similarities among mentees who shared the same mentor. We have now included these analyses on p. 3 of the Supplementary Materials.

Variable	ICC for Mentor Cluster	Variance of Mentor Cluster
Anxiety	0.08	$B = 0.12, p = .251$
Belonging	0.06	$B = 0.08, p = .481$
Confidence	0.06	$B = 0.08, p = .475$
Motivation	0.06	$B = 0.05, p = .493$
Graduate Intentions	0.05	$B = 0.11, p = .724$
Career Intentions	0.06	$B = 0.10, p = .449$
Emotional Well-being	0.08	$B = 0.15, p = .547$

2. I had trouble understanding the analysis strategy (e.g., what all of the random effects were), and this might be because I am not familiar with multilevel modeling in MPlus. Given that not all readers will be familiar with MPlus, I suggest that the authors include equations to represent their models as these equations will be consistent across analysis software. One recommendation is to make sure that the equations presented are the ones used to generate the results. For example, it does not seem that the dummy codes presented were the ones used to test the A and B analyses (A: changes over time per condition and B: whether change in each condition is different from change in other conditions) listed in the first paragraph on page 10. If they were, then it was not clear how because the dummy codes seem to test the difference between having a female mentor and no mentor (first code) and then the difference between having a male mentor and no mentor (second code). I think presenting equations might be useful in clarifying exactly what analysis was done to produce what results, but other strategies may also work.

Thanks for the suggestion. We now include equations for the multilevel analyses on p. 12. In each model, the reference slope (no mentor condition) is estimated, as well as the difference in slopes between the no mentor condition and the other two mentor conditions. The slopes for the male and female conditions are calculated by adding this slope difference to the reference slope. Data analysis syntax is included in the OSF; we point interested reviewers to the OSF folder with syntax on p. 29.

3. Dummy codes: It would be useful to note that the reference group is the no mentor condition.

We now clarify that the reference group is the no mentor condition on p. 11.

4. Longitudinal outcomes:

a. I was expecting the authors to first report omnibus tests that showed whether the mentor condition significantly interacted with time to predict any of the outcomes as a way to control for Type I error rate. Then, I would expect them to do the A and B analysis strategies they listed on page 10. Is there a reason they did not conduct omnibus tests like this? Did they control for Type I error rate in a different way?

Since our data is nested, they violate independence assumptions required in omnibus ANOVA and ordinary least-squares (OLS) multiple regression analyses. Thus, by using multilevel analyses, where we account for the fact that our data are nested within individuals, we are avoiding potential Type I errors and biased estimates that may emerge in ANOVA and OLS multiple regression (see Peugh, 2010). That is why we did not conduct omnibus tests.

b. In addition to understanding changes over time, it would also be useful to know at what timepoints the outcomes differed between conditions. For example, was anxiety (relative to motivation) significantly lower for those with female mentors, relative to those with male mentors or no mentors, at the final timepoint of the study? Was it significantly different at any other timepoints as well? Knowing the answers to these questions would be useful in understanding how long it takes for this mentor intervention to yield effects.

Our focus for the study was on changes over time, rather than mean differences. Given that follow-up surveys were taken over a variable period of time (i.e., winter to late summer in order to maximize response rate), we believe that capturing participants' changes using their real survey timepoints rather than aggregating within each year would be the most accurate representation of our data. However, in our updated figures, we now report standard error bars of each condition's estimated (using FIML) mean at each year, which should give readers a sense of when condition differences begin to emerge.

c. Did the authors consider any nonlinear changes over time? Some interventions show significant effects early on which remain steady over time but do not increase, which would result in a nonlinear trajectory. Did the authors conduct any analyses to examine whether nonlinear trajectories might be more appropriate?

We conducted quadratic analyses in response to Reviewer 3’s question. In each analysis, there was either none or one quadratic effect; but the patterns were inconsistent: sometimes it emerged in the no mentor condition while at other times in the male mentor condition. Furthermore, multiple quadratic models did not fit the data any better than the linear growth curve models. In order to be consistent across dependent variables and not add additional complexity to our analyses using a relatively small sample size, we elected to use linear growth curve models for all continuous variables.

5. Figures: Can the authors include standard error or confidence interval bands in the figures? It would be interesting to know more about the variability associated with outcomes for people with female vs. male mentors. It might speak to whether some male mentors are as helpful as female mentors. If this is the case, it might ultimately be worthwhile to try to understand what factors allow male mentors to be helpful as well—given that women in engineering are likely to encounter more men at “mentor” levels than women at these levels.

Thanks for the suggestion. In each figure, our lines now include standard error bars.

Minor Methodological Questions

1. I’m curious about the distribution of the internship measure and why the authors decided to make this a binary measure. Were there many students who had more than one internship? If so, might there be value in using this as a “true” count measure, where all internships are counted?

We elected to use this as a dichotomous measure for two reasons. First, the number of participants who completed the survey varied from year to year, which means that a non-response in a given year would count as zero internships for that year if we used a count measure, which might not be accurate. Second, there was variability in the number of years it took participants to graduate from college (between 3-5 years), which would alter the number of time points and internships that they could have had in college. For both these reasons we treated internships as a dichotomous variable not a continuous one. We now specify this on p. 27.

2. I found it a little difficult to understand what percentage of students completed measures at each timepoint of the study. I suggest the authors include a table or figure that shows what percentage of students across each mentor condition participated at each timepoint.

In the revised paper, we provide a data table (see Table 1 on p. 38; also inserted below) that shows the number and percentage of participants who completed a survey at each time point as well as our percentage of participants for whom we had data for their major at graduation.

	Year 1 (Baseline: Before Intervention)	Year 1 (Middle of Intervention)	Year 1 (End of Intervention)	Year 2	Year 3	Year 4	Year 5+
Number (percent) of survey respondents	150 (100%)	150 (100%)	150 (100%)	102 (68%)	49 (33%)	61 (41%)	103 (69%)

Number (percent) of
transcripts collected

-

-

-

-

-

-

150 (100%)

3. Related to the prior point, are the data from the survey after graduation included here? The figures go up to year 4, so it seems like they aren't. If not, why not? Were the measures the same?

We did collect participant data one year after college graduation. These had not been included in our original manuscript. Based on Reviewer 3's comment, we now include all post-graduation data in the revised manuscript. The measures remained the same.

Minor Questions about the Discussion

1. I was surprised the authors did not address reasons for why the mentor intervention is effective, other than stereotype inoculation. For example, is it possible that female mentors behave in different ways? Is it possible they provide useful information or a perspective that male mentors don't?

We thank Reviewer 3 for this important suggestion. We offer two theory-driven mechanisms to explain why female mentors were more effective than male mentors (now included in the Discussion on pp. 19-20). First, sociological research shows that for students who are minorities in a given field, developing a strong social network of people who share their marginalized identity in the target profession is especially helpful for professional success. In one study, Yang et al. (2019) found that female students in MBA programs were more successful in getting high-ranked jobs after graduation if they had a close network of women who advised them about where to apply, how to interview, and offered other background information about companies. Advice from close female connections (vs. male connections) may be especially effective because the women giving informal advice are more cognizant of potential barriers about which male peers may not be aware. In general, access to high-value social networks gives people an advantage in finding employment; people who are minoritized in a profession often have less access to professional connections that open doors (Pedulla & Pager, 2019; Trimble & Kmec, 2011). By connecting participants with female mentors in the first year of college, we may have provided new students their first connection in engineering. They may have used this relationship to expand their social network in engineering across the 3-5 years of college. The connections made through the female mentors (as compared to the male mentors) may have been more valuable for women mentees' success, similar to Yang et al. (2019). Our finding that women with female mentors secured more engineering internships in college suggests that perhaps female mentors connected their mentees to recruiters and hiring managers with whom they had relationships. Future research should directly test if same-sex mentoring relationships expands mentees' access to high-value social networks.

A second potential mechanism may be that female mentors elicited a change in their mentees' mindset in a critical transitional period of their life. As theorized by Walton (2014), interventions may have a long-term impact if they change "recursive processes" early on (p. 80). If an intervention is able to alter individuals' way of thinking and help them gain more constructive

mindsets regarding their learning in the beginning of a new environment, they will be more likely to persist in their education. In contrast, if students remain caught in cycles of psychological threat and poor performance, they will be less likely to persist. Applying mindset change to our research, the benefit of female (as compared to male) mentors may have persisted because women students were able to develop constructive mindsets regarding engineering and develop their professional identity as a future engineer through support from same-sex mentors and by envisioning those mentors as their future self. Future research should directly examine these and other potential mechanisms driving the success of same-sex mentorship interventions that have long-lasting effects.

2. Did belongingness not mediate the relationship between mentorship and academic outcomes here? The discussion (pages 17 to 18) seems to imply that it did not but I was not sure where that was reported.

In the current manuscript, belonging did not significantly differ between mentor conditions (see pp. 1-2 of Supplementary Materials); thus, it did not mediate the relationship between mentorship and academic outcomes. We were referring to our 2017 paper, which we cited (p. 21), in which we found that in the first year of college (only), belonging significantly mediated the effect of female mentors (vs. the other two conditions) on students' intentions to pursue an engineering career.

3. I would be careful about saying that assignment to a male mentor or no mentor "increased" women's anxiety (bottom of p. 16). It's more likely that the "baseline" is an increase in anxiety over the course of college, and female mentors simply protect against the occurrence of this, rather than male mentors or no mentors causing greater anxiety.

Yes, that's a good cautionary point. We've changed this language. We are now careful to consistently describe our results as showing that female mentors provide a protection effect.

REVIEWER COMMENTS

Reviewer #2 (Remarks to the Author):

Ms. No.: NCOMMS-21-21677A

Title: The enduring impact of female peer mentors in the first year of college on women students in engineering through college to post-graduation

The authors did a good job responding to the issues I noted in my first review. However, there are a few lingering concerns.

On page 10, the authors suggest that they identified “the psychological mechanism that helps explain why female mentorship is effective.” Of course, there may be additional mechanisms beyond the one they identified (see Memon et al 2018) and this needs to be clarified. Though, they do speculate about other mechanisms in the discussion so they do recognize that they did not identify THE mechanism; they just need to be consistent about it.

Related to mechanism, I am confused about the differences between their constructs of confidence versus motivation when looking at the items. Did the authors do factor analyses on the confidence and motivation items? Are they indeed two separate factors? The motivation items read like self-efficacy, or confidence, items. How is the item “I have the skills and abilities to be successful in my engineering related classes” not a measure of self-efficacy? Or, “I feel confident about my engineering related classes this year.”

The authors note that they collected information regarding a summary of topics discussed in mentoring meetings. They might consider coding these notes to look for similarities and differences across the female and male mentor meetings that might give more insight into mechanism.

The reason that I asked about whether there was any information about whether the female and male mentors differed in terms of having internships in my first review is because it is important to in this work to demonstrate that these groups of mentors do not differ in important ways that are confounded with gender and could potentially explain some of the differences in outcomes. The authors should more clearly discuss the limitations associated with the potential of there being other differences between the groups that are not related to their purported mechanism(s).

Although I do think there are some lingering issues that need to be addressed, I remain positive about this manuscript.

Reviewer #2 (Remarks to the Author):

Ms. No.: NCOMMS-21-21677A

Title: The enduring impact of female peer mentors in the first year of college on women students in engineering through college to post-graduation

The authors did a good job responding to the issues I noted in my first review. However, there are a few lingering concerns.

On page 10, the authors suggest that they identified “the psychological mechanism that helps explain why female mentorship is effective.” Of course, there may be additional mechanisms beyond the one they identified (see Memon et al 2018) and this needs to be clarified. Though, they do speculate about other mechanisms in the discussion so they do recognize that they did not identify THE mechanism; they just need to be consistent about it.

Thank you for this clarification. In our updated manuscript on p. 10 and p. 19, we removed the language stating that we identified *the* singular mechanism. We now clarify that among our measured variables, we identify which of these serves as *a* psychological mechanism underlying these differences.

Related to mechanism, I am confused about the differences between their constructs of confidence versus motivation when looking at the items. Did the authors do factor analyses on the confidence and motivation items? Are they indeed two separate factors? The motivation items read like self-efficacy, or confidence, items. How is the item “I have the skills and abilities to be successful in my engineering related classes” not a measure of self-efficacy? Or, “I feel confident about my engineering related classes this year.”

We did not conduct factor analyses on these two particular items; these items were drawn from separate concepts in prior research. Motivation was drawn from threat and challenge (e.g., Major & Seery, 2001; Mendes et al., 2001; Moore et al., 2012). While threat refers to the motivation to withdraw because the demands of a situation overwhelm one’s inner resources, challenge is the motivation to approach when one’s inner resources are greater than situational demands. Because the term “challenge” is social psychological jargon whose meaning may be ambiguous to readers outside of psychology, we relabeled “challenge” as “motivation” to make it more accessible to a broad readership. Our measure of confidence, however, was drawn from literature on perceived self-efficacy (e.g., Bandura & Wessels, 1994; Stout et al., 2011), which measures one’s confidence in their own abilities.

Furthermore, an important distinction between our measure of motivation and confidence is that all items for motivation measure participants’ experiences in their *engineering classes specifically*, while all the items for confidence measure their self-perceived *global ability* in engineering. We now specify this distinction throughout the manuscript. Given that our constructs target different contexts (local classes vs. global discipline) and items are drawn from prior research and theory, we elected to keep these constructs separate.

The authors note that they collected information regarding a summary of topics discussed in mentoring meetings. They might consider coding these notes to look for similarities and differences across the female and male mentor meetings that might give more insight into

mechanism.

Mentor reports of each meeting were coded for whether the meeting was primarily social or professional. The percentage of meetings that were primarily social or professional did not significantly differ by mentor sex ($ps > .58$). That said, mentor reports were cursory and not sufficiently detailed to pick up nuances in mentor-mentee conversations. Future research would benefit from measuring and analyzing conversations between mentors and mentees in more granular form and assess if they differentially predict mentee outcomes.

The reason that I asked about whether there was any information about whether the female and male mentors differed in terms of having internships in my first review is because it is important to in this work to demonstrate that these groups of mentors do not differ in important ways that are confounded with gender and could potentially explain some of the differences in outcomes. The authors should more clearly discuss the limitations associated with the potential of there being other differences between the groups that are not related to their purported mechanism(s). Thank you for this important point. We now highlight on p. 21 that a limitation of the current study is our inability to assess whether mentors who were more successful getting internships promoted their mentees' success with internships as well. We note that future research should examine mentor outcomes and whether the beneficial effects of female mentorship remain while accounting for mentor success.

Although I do think there are some lingering issues that need to be addressed, I remain positive about this manuscript.

Thank you for your thoughtful comments and questions about our manuscript.